# Adapt Data to Model: Adaptive Transformation Optimization for Domain-shared Time Series Foundation Models

**Yunzhong Qiu**[*], **Zhiyao Cen**[*], **Zhongyi Pei**[†], **Chen Wang, Jianmin Wang**
School of Software, BNRist, Tsinghua University, China
{qiuyz24,cenzy23}@mails.tsinghua.edu.cn
{peizhyi,wang_chen,jimwang}@tsinghua.edu.cn

## Abstract

Large time series models (LTMs) have emerged as powerful tools for universal forecasting, yet they often struggle with the inherent diversity and nonstationarity of real-world time series data, leading to an unsatisfactory trade-off between forecasting accuracy and generalization. Rather than continually fine-tuning new LTM instances for each domain, we propose a data-centric framework, time-series adaptive transformation optimization (TATO), that enables a single frozen pre-trained LTM to adapt to diverse downstream domains through an optimally configured transformation pipeline. Specifically, TATO constructs three representative types of transformations, including context slicing, scale normalization, and outlier correction, to help LTMs better align with target domain characteristics. To ensure robustness, we incorporate carefully selected time series augmentations and a two-stage ranking mechanism that filters out pipelines underperforming on specific metrics. Extensive experiments on state-of-the-art LTMs and widely used datasets demonstrate that TATO consistently and significantly improves domain-adaptive forecasting performance, achieving a maximum reduction in MSE of 65.4% and an average reduction of 13.6%. Moreover, TATO is highly efficient, typically completing optimization in under 2 minutes, making it practical for real-world deployment. The source code is available at https://github.com/thulab/TATO.

## 1 Introduction

With the rapid advancement of large language models (LLMs), there has been growing interest in extending the capabilities of large models to time series forecasting, i.e., *large time series models* (LTMs) (Liu et al., 2024b). LTMs are designed to handle multiple downstream tasks using a single foundational model. Two main technical directions have been widely adopted. The first involves building time series native models pre-trained on large-scale time series data (Liu et al., 2024b; Fatir Ansari et al., 2024; Woo et al., 2024; Garza & Mergenthaler-Canseco, 2023). The second adapts large language models (LLMs) pre-trained on diverse text data to time series analysis tasks (Gruver et al., 2024a; Zhou et al., 2023; Liu et al., 2024a). These large models demonstrate significantly better generalization than traditional models, effectively supporting few-shot learning, in-context learning, and even zero-shot inference.

This capability, known as zero-shot forecasting (Gruver et al., 2024b), is a cutting-edge paradigm that enables accurate predictions on unseen data without additional training, much like using Chat-GPT. However, LTMs face significantly greater challenges than LLMs in achieving effective zero-shot forecasting for time series. The fundamental issue lies in the inherent diversity of time series data—different domains exhibit distinct temporal characteristics, making it extremely difficult for a single LTM to generalize across all domains. A straightforward approach to improving domain-

---

[*]Equal contribution
[†]Corresponding author

specific performance is to finetune pretrained LTMs, but this compromises the model's generalizability and incurs prohibitive computational costs as the number of target domains increases.

To address this challenge, we propose a novel paradigm for applying LTMs that adapts data to a shared foundation model for different domains. This approach strikes a balance between achieving strong performance across diverse domains and minimizing training costs through efficient data preprocessing, enabling a single LTM to serve all domains effectively. We term this paradigm *Frozen Foundation Model-based Domain-Shared Forecasting (FrozenForecasting)*, in which the foundation model remains frozen while concurrent lightweight data adaptations are allowed to handle diverse downstream domains.

The focus on model-centric development has long overlooked the potential of adapting the data itself—a perspective that our work seeks to revitalize. For decades, deep learning has been dominated by an end-to-end paradigm, where researchers have primarily focused on pretraining or finetuning models to improve performance. This trend has continued in recent work on LTMs, leaving the value of data transformation underexplored. To illustrate the impact of time series transformations on LTMs, we present three motivating examples that compare original LTM predictions with those with optimized transformations in Figure 1.

Each example, on a distinct LTM and a typical dataset, illustrates a representative case in which the transformation has a significant impact. In Example 1, Moirai (Woo et al., 2024) produces noisy predictions due to limited input context. By extending the input and applying downsampling to emphasize periodicity, the predictions become significantly more stable. In Example 2, Timer (Liu et al., 2024b) initially misinterprets outliers as periodic patterns. After applying $k$-sigma outlier detection and linear interpolation, the predictions align closely with the ground truth trend. In Example 3, Chronos (Fatir Ansari et al., 2024) generates conservative predictions that regress toward the input mean. Applying differencing induces stationarity, enabling the model to better capture overall trends. These examples demonstrate that targeted data transformations can substantially enhance LTM performance, motivating the need for automated transformation optimization in FrozenForecasting.

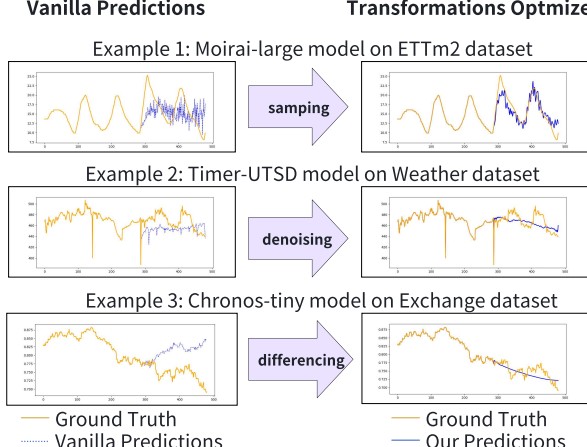

Figure 1: Three examples illustrating how data transformations enhance LTM predictions. (a) Downsampling stabilizes noisy Moirai predictions on ETTm2. (b) Outlier detection and interpolation correct Timer's misinterpretation of anomalies on Weather. (c) Differencing enables Chronos to capture trends on Exchange by inducing stationarity. They demonstrate the potential of transformation optimization for FrozenForecasting.

To enable effective FrozenForecasting, we introduce an automated framework that optimizes time series transformations for LTMs. The proposed approach formulates transformation discovery as a hyperparameter optimization problem to search for the most effective preprocessing pipeline. To ensure robustness, we incorporate data augmentation during large-scale search. Each optimization trial selects and applies a combination of representative transformation operators—such as context slicing, scale normalization, and outlier correction—to the input time series. A two-stage evaluation mechanism is then employed to identify the optimal transformation pipeline, tailored specifically for FrozenForecasting with LTMs. In summary, this paper makes the following major contributions:

- **New Paradigm for LTMs.** We introduce FrozenForecasting, a novel paradigm that addresses the practical requirement of deploying large time series models (LTMs): a single frozen model that generalizes across all domains. To enable this, we propose TATO (Time-series Adaptive Transformation Optimization), a preprocessing-based framework that provides a universally beneficial, model-agnostic solution for enhancing frozen LTM performance across diverse domains.

- **Specialized Transformation Search Space.** We carefully curate a comprehensive set of time series transformations, forming a compact yet expressive search space for TATO to discover optimal preprocessing pipelines. This space encompasses three categories of performance-critical operators: context slicing, scale normalization, and outlier correction, each designed to mitigate domain-specific data characteristics.

- **Remarkable Empirical Results.** Through extensive experiments on eight popular time series datasets, we demonstrate that TATO consistently and significantly improves the performance of several state-of-the-art LTMs. Our approach achieves a maximum mean squared error (MSE) reduction of 65.4% and an average reduction of 13.6%, while maintaining exceptional efficiency, with optimization typically completed in under 2 minutes.

## 2 RELATED WORK

### 2.1 LARGE TIME SERIES MODELS

Unlike traditional models such as Autoformer (Wu et al., 2021) and PatchTST (Nie et al., 2022), which are typically trained and applied to specific datasets, LTMs are designed as foundational models capable of zero-shot forecasting. Some LTMs are trained from scratch on large-scale time series datasets. For instance, TimeGPT (Garza & Mergenthaler-Canseco, 2023) claims to be the first time series foundation model capable of handling multiple tasks on datasets with a single model. Timer (Liu et al., 2024b) trained a GPT-style architecture on the constructed Unified Time Series Dataset (UTSD) and demonstrated notable feasibility and generalizability across various tasks. Moirai (Woo et al., 2024), trained on the Large-scale Open Time Series Archive (LOTSA), excels in zero-shot forecasting and handles multivariate time series. TimesFM (Das et al., 2023) uses a decoder-style attention mechanism with input patching for forecasting tasks. Other LTMs typically leverage existing LLMs and adapt them to time series tasks. For example, Chronos (Fatir Ansari et al., 2024), finetuned from LLMs, tokenizes time series data using a fixed vocabulary. AutoTimes (Liu et al., 2024a) leverages LLMs to align time series with natural language. However, given the extremely high diversity and non-stationarity of time series downstream domains, the generalizability of LTMs is severely limited, as shown in Figure 1. Finetuning is a practical approach to applying LTMs to downstream domains. It creates new, targeted LTMs based on target data, thereby increasing the number of LTM instances. Instead, TATO modifies how LTMs process data, transforming it to improve FrozenForecasting performance via automated transformation pipelines. Compared to finetuning, TATO offers a novel paradigm for balancing specificity and generalization in LTMs, enhancing specificity without compromising generalizability.

### 2.2 TIME SERIES TRANSFORMATION

Time series transformations (Tawakuli et al., 2024) are crucial for enhancing data quality and improving model performance, particularly in training or data mining applications. Normalization techniques, such as the Z-score (Alexandropoulos et al., 2019) and P-norm (Zheng & Casari, 2018), are widely used in machine learning to rescale numeric features, producing values with similar ranges. Data cleaning addresses data quality issues by identifying and managing anomalies and outliers, which can negatively impact model performance. The interquartile range (IQR) (Wang et al., 2018) and the local outlier factor (LOF) (Breunig et al., 2000) are widely used methods for identifying and handling outliers. Sensor fusion techniques like the Kalman Filter (KF) (Julier & Uhlmann, 1997), discretization methods such as K-means clustering (Gupta et al., 2010), and feature extraction methods like principal component analysis (PCA) (Abdi & Williams, 2010) all serve to integrate, summarize, and reduce the dimensionality of time series data. The above methods provide ways to handle the variety of time series, which can be used as operators within the TATO framework.

To enhance sample diversity during transformation optimization, TATO leverages a range of data augmentations commonly used in time series analysis (Iglesias et al., 2023; Javeri et al., 2021). These include flipping augmentations (magnitude flip and time flip) that reverse sequences or magnitudes to create new variations (Um et al., 2017; Fawaz et al., 2018), warping augmentations (magnitude warp and time warp) that distort data to simulate different conditions (Um et al., 2017; Park et al., 2019; Jeong et al., 2021), noise augmentations (EWMA smoothing and jitter) that improve data resilience (Flores et al., 2021; Rashid & Louis, 2019), and translation that shifts data horizontally

or vertically to simulate distribution shifts (Huang et al., 2017). Importantly, these augmentations are applied exclusively during the optimization phase to enrich the candidate pool; the final selected transformation pipeline for inference does not include any augmentation.

## 2.3 Test-Time Adaptation

Test-time adaptation (TTA) techniques address performance degradation caused by distribution shifts between training and test data by adapting models during inference (Liang et al., 2024). Traditional TTA approaches typically involve self-supervised training of neural networks on unlabeled test data (Zhang et al., 2022; Schneider et al., 2020; Iwasawa & Matsuo, 2021; Boudiaf et al., 2022). More recently, transformation-based methods have emerged for time series models. TokenMerging (Götz et al., 2024) applies preprocessing to reduce token count in transformer-based time series models, improving inference throughput. NewNorm (Kim et al., 2024) addresses distribution shifts through trend estimation and self-supervised learning of new normalities. While these methods target specific challenges, TATO takes a broader approach by systematically optimizing a series of transformations to enhance frozen LTM performance across diverse domains—a core requirement of the FrozenForecasting paradigm.

## 3 Method

### 3.1 The Paradigm of Adapting Data to Model

To address the practical need to deploy LTMs across diverse domains, we introduce FrozenForecasting, a paradigm that performs domain-shared forecasting using a single frozen LTM. Within this paradigm, we propose TATO, an automated framework that adapts data rather than models, optimizing time series transformations to enhance frozen LTM performance while avoiding costly, sometimes risky finetuning.

**definition 1** (Frozen Foundation Model-based Domain-shared Forecasting). *A forecasting paradigm in which a single pre-trained foundation model remains frozen during inference, while lightweight, domain-specific adaptations are applied to input data to enable accurate forecasting across diverse downstream domains.*

To enable FrozenForecasting with existing LTMs, we propose TATO (Time-series Adaptive Transformation Optimization), a framework that systematically discovers and applies optimal transformation pipelines to improve forecasting performance across diverse domains while the model itself remains frozen. Let $D$ be a domain of time series, from which two subsets $D_{\text{history}}$ and $D_{\text{future}}$ are sampled. $\mathcal{H}$ denotes a configuration space for time series transformation pipelines and $h$ denotes one instance of the space. Given a large time series model $M$, time series adaptive transformation optimization for FrozenForecasting (TATO) can be formulated as follows:

**definition 2** (Time-series Adaptive Transformation Optimization for FrozenForecasting).

$$h^* = \min_{h \in \mathcal{H}} \mathcal{L}(M, D_{history}, h)$$

*where $\mathcal{L}(M, D_{history}, h)$ denotes the loss that the model $M$ achieves when the sampled data $D_{history}$ are transformed by $h$. It should be noted that the optimal transformation pipeline $h^*$ is searched on $D_{history}$, while $D_{future}$ is used for final testing.*

While TATO leverages historical data, it differs fundamentally from finetuning in several key aspects. First, the LTM remains completely frozen—only the data transformation pipelines are optimized, leaving model parameters untouched. This enables a single model to adapt to diverse domain distributions without the one-to-one mapping required by finetuning. Second, TATO is substantially more efficient, requiring far less historical data. In our experiments, just 500 samples are sufficient for TATO to achieve considerable improvements across various scenarios. To safeguard against data shift during optimization, we carefully selected diverse time series augmentations that maintain sufficient variety while preserving each sample's intrinsic characteristics. As a result, transformation pipeline optimization carries a significantly lower risk of overfitting than finetuning, underpinning TATO's universal effectiveness.

## 3.2 THE TATO FRAMEWORK

The TATO framework operates through three core stages: data preparation, transformation pipeline optimization, and optimal pipeline selection, as shown in Figure 2.

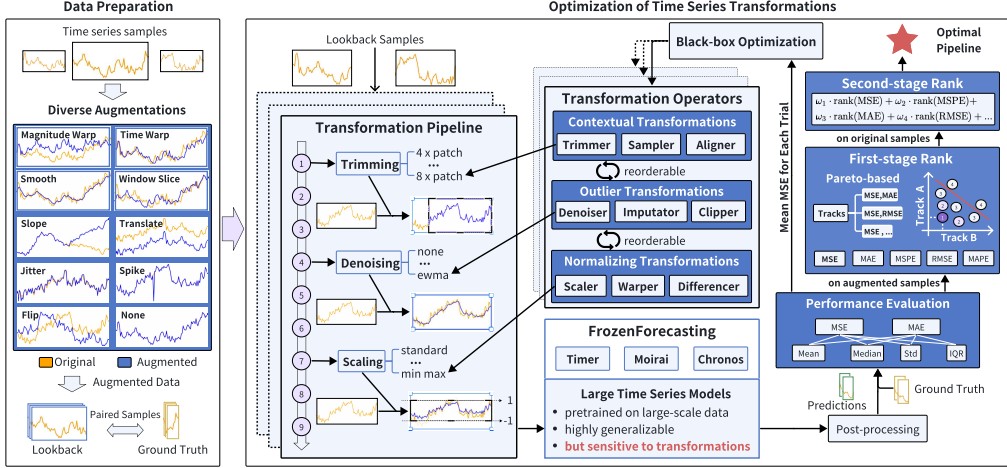

Figure 2: Overview of the TATO framework. The framework consists of three main stages: (1) Data preparation, where diverse augmentations are applied to input samples to improve robustness; (2) Optimization of time series transformations, where a black-box optimizer searches for effective transformation pipelines comprising various preprocessing operators (e.g., trimming, normalization, denoising); and (3) Two-stage pipeline selection, where candidate pipelines are first filtered via Pareto ranking on validation metrics, followed by weighted multi-indicator ranking to select the optimal transformation pipeline for frozen LTM forecasting.

### 3.2.1 DATA PREPARATION

As defined in Definition 2, the optimal transformation pipeline $h^*$ is identified from historical samples but tested on unseen data. To address potential discrepancies between $D_{\text{history}}$ and $D_{\text{future}}$, we aim to maximize $D_{\text{history}}$'s diversity by incorporating a range of augmentation methods. The augmentation methods used in TATO are carefully selected from the literature (Iglesias et al., 2023; Javeri et al., 2021; Zhang et al., 2022), including both commonly used methods and those designed to generate samples that are vastly different (e.g., spikes, slopes) and may occur in real-world scenarios. Flipping augmentations, including magnitude and time flip, reverse the sequence or magnitude of data points to create new variations (Um et al., 2017; Fawaz et al., 2018). Warping augmentations, such as *magnitude warp* and *time warp*, distort data to simulate different conditions (Um et al., 2017; Park et al., 2019; Jeong et al., 2021). Noise injections include *exponentially weighted moving average (EWMA)* for smoothing and *jitter* for adding random noise (Flores et al., 2021; Rashid & Louis, 2019). *Translation* shifts data horizontally or vertically to simulate distribution shifts, addressing mean offsets in test data (Huang et al., 2017). Adding *slopes* (Wen & Keyes, 2019) can simulate datasets with underlying trends.

### 3.2.2 TRANSFORMATION PIPELINE OPTIMIZATION

The transformation pipelines consist of nine operators categorized into three types: (1) contextual transformations that slice the inference context, (2) normalization transformations that adjust the value range of inputs and outputs, and (3) outlier transformations that mitigate the risk of performance degradation due to outliers.

All data transformations consist of a pair of pre-process and post-process operators applied before the input to LTMs and after the output from LTMs. Such operators are predefined by a set of hyperparameters to control the implemented data transformation. They are optimized by hyperparameter optimization techniques, such as the widely used Bayesian optimization algorithm *Tree-structured*

*Parzen Estimator (TPE)* (Bergstra et al., 2011; Watanabe, 2023). During the specific optimization process, the data transformation order is also considered using rule-based heuristics and search-tree branch-cutting to control overall computational complexity. After a fixed number of trials, the optimal data transformation pipelines are identified using Pareto-based ranking across multiple assessment metrics to ensure robust selection. The technical details of data transformations are presented in Appendix A.3

To prevent inappropriate or repeated trials caused by meaningless reordering, we predefine three orders as candidates rather than searching all possible orders. The predefined transformations are ordered according to several heuristic rules. First, regarding efficiency, the operator *Trimmer* is placed at the head because it may reduce the sample length for subsequent operators. Second, outlier transformations should generally precede normalizing transformations, as outlier values can significantly affect normalization. Third, some operators, such as *Aligner*, prefer to be placed in the tail to avoid introducing interference to others.

### 3.2.3 TWO-STAGE PARETO-BASED RANKING OF PERFORMANCE

To quantitatively evaluate forecasting performance, we compute multiple error metrics on the transformed samples, including mean squared error (MSE), mean absolute error (MAE), root mean squared error (RMSE), mean absolute percentage error (MAPE), and mean squared percentage error (MSPE). For enhanced robustness, we also consider their summary statistics—median, standard deviation (STD), and interquartile range (IQR). A key challenge is that no single trial consistently outperforms the others across all metrics, making averaging across metrics a risk of elevating poorly performing trials. To address this, we adopt a two-stage Pareto-based ranking approach (Palakonda & Mallipeddi, 2017) for selecting the final transformation pipeline.

In the first stage, we filter out risky trials that underperform on any metric subset across all augmented samples. Specifically, we construct a Pareto set in which each retained trial must not rank poorly on any given metric combination across the augmented data, thereby eliminating trials that excel only on specific metrics. For efficiency, we set the filtering threshold to retain 16 candidates and exclude the remaining trials for each metric subset. In the second stage, we rank the candidate transformation pipelines using only the original samples, not the augmented ones. The ranking is computed as a weighted sum of all metric ranks, with higher weights assigned to MSE and MSPE. The top-ranked transformation pipeline is then applied to test data for FrozenForecasting. Further details are provided in Appendix A.4.

## 4 EXPERIMENTS

### 4.1 DATASETS AND BASELINES

Five major datasets are included in our experimental analysis: ETT (4 subsets), Electricity, Exchange, Traffic, and Weather. Specifically, we follow the same data preparation and train-validation-test set split protocol used in TimesNet (Wu et al., 2022). Only 2% or less of the total training samples are used in TATO, and the test set is consistently held out during optimization of the transformation pipeline. Several advanced transformer-based LTMs are introduced in our assessment. Timer(Liu et al., 2024b) employs a GPT-style architecture and benefits from extensive pre-training, enabling it to autoregressively predict the next time series token in a unified generative manner. Moirai(Woo et al., 2024), a masked encoder-based universal time series forecasting transformer trained on LOTSA, demonstrates superior performance as a zero-shot forecaster compared to full-shot models. Chronos(Fatir Ansari et al., 2024) tokenizes time series values using scaling and quantization into a fixed vocabulary and trains existing transformer-based language model architectures on these tokenized time series using cross-entropy loss.

### 4.2 EVALUATION METHOD

To comprehensively evaluate the effectiveness of TATO, we employ a set of error metrics and their statistical measures to assess the relative performance improvements achieved through our data transformations. Mean Squared Error (MSE) and Mean Absolute Error (MAE) are used to quantify the performance of the time series forecasting. The distributional properties of the metrics, such as

the *mean* and *median*, are computed to provide a robust measure of central tendency and to reduce the influence of outliers. We also used the standard deviation (STD) to measure the spread around the mean and the interquartile range (IQR) to represent the middle half of the error values, providing additional perspectives for evaluating robustness.

To assess the effectiveness of optimized data transformations, we compare the error metrics with and without applying TATO. Let $e_V$ be the error metric of the vanilla baselines, following the transformations from the LTMs and a lookback length of $7 \times 96$ from Timer(Liu et al., 2024b). Let $e_T$ be the error metric using optimized transformations. We define a metric, Relative Percentage Promotion in Error, as follows:

**definition 3** (Relative Percentage Promotion in Error)**.** *Relative Percentage Promotion in Error(%Promotion) from baseline vanilla to our method TATO is calculated as* $\%Promotion = \frac{e_V - e_T}{e_V}$. *A positive %Promotion value indicates performance enhancement.*

## 4.3 EFFECTIVENESS

We evaluate TATO's effectiveness by reporting the relative percentage improvement in error (%Promotion) achieved over vanilla frozen LTMs across various models, datasets, and prediction horizons in Table 1. The experiments cover 192 distinct scenarios (8 datasets × 4 prediction lengths × 8 LTMs), demonstrating the universality of our approach. For robustness, each experiment is repeated three times. TATO uniformly samples 500 training instances per dataset to optimize transformation pipelines, with a budget of up to 500 trials per scenario.

Table 1: MSE and MAE reduction achieved by TATO across different models and datasets. Results are averaged over prediction horizons 24, 48, 96, 192. Positive %Promotion, in bold, indicates performance improvement (error reduction) achieved by TATO compared to the baseline frozen LTM. Best results in each row are highlighted in red, worst in blue.

| Models | | Timer-UTSD | | Timer-LOTSA | | Moirai-small | | Moirai-base | | Moirai-large | | Chronos-tiny | | Average | |
|---|---|---|---|---|---|---|---|---|---|---|---|---|---|---|---|---|
| **Error Metric** | | **MSE** | **MAE** | **MSE** | **MAE** | **MSE** | **MAE** | **MSE** | **MAE** | **MSE** | **MAE** | **MSE** | **MAE** | **MSE** | **MAE** |
| ETTh1 | Vanilla | 0.4096 | 0.4846 | 0.4372 | 0.5010 | 0.4628 | 0.5163 | 0.4901 | 0.5249 | 0.4381 | 0.4989 | 0.4371 | 0.5055 | 0.4458 | 0.5052 |
| | with TATO | 0.3901 | 0.4762 | 0.4379 | 0.5010 | 0.4460 | 0.5044 | 0.4503 | 0.5081 | 0.4141 | 0.4823 | 0.4460 | 0.5036 | 0.4307 | 0.4960 |
| | **%Promotion** | **4.8%** | **1.7%** | -0.2% | 0.0% | **3.6%** | **2.3%** | **8.1%** | **3.2%** | **5.5%** | **3.3%** | -2.0% | **0.4%** | **3.4%** | **1.8%** |
| ETTh2 | Vanilla | 0.3258 | 0.4413 | 0.3646 | 0.4777 | 0.5249 | 0.5660 | 0.4657 | 0.5287 | 0.4494 | 0.5119 | 0.4181 | 0.4892 | 0.4247 | 0.5025 |
| | with TATO | 0.3203 | 0.4375 | 0.3431 | 0.4535 | 0.4691 | 0.5344 | 0.4538 | 0.5211 | 0.3688 | 0.4692 | 0.4297 | 0.5001 | 0.3975 | 0.4860 |
| | **%Promotion** | **1.7%** | **0.9%** | **5.9%** | **5.1%** | **10.6%** | **5.6%** | **2.6%** | **1.4%** | **17.9%** | **8.3%** | -2.8% | -2.2% | **6.4%** | **3.3%** |
| ETTm1 | Vanilla | 0.2963 | 0.3979 | 0.3687 | 0.4550 | 0.2554 | 0.3682 | 0.2648 | 0.3701 | 0.2380 | 0.3549 | 0.2815 | 0.3831 | 0.2841 | 0.3882 |
| | with TATO | 0.2890 | 0.3910 | 0.2189 | 0.3380 | 0.2259 | 0.3467 | 0.2286 | 0.3445 | 0.2253 | 0.3445 | 0.2457 | 0.3561 | 0.2389 | 0.3540 |
| | **%Promotion** | **2.5%** | **1.8%** | **40.6%** | **25.7%** | **11.6%** | **5.8%** | **13.7%** | **6.0%** | **5.3%** | **2.9%** | **12.7%** | **7.0%** | **15.9%** | **8.8%** |
| ETTm2 | Vanilla | 0.4644 | 0.5224 | 0.3146 | 0.4166 | 0.2969 | 0.3996 | 0.3747 | 0.4391 | 0.3288 | 0.3984 | 0.4006 | 0.4307 | 0.3633 | 0.4345 |
| | with TATO | 0.4368 | 0.4909 | 0.2506 | 0.3520 | 0.2374 | 0.3405 | 0.2512 | 0.3477 | 0.2277 | 0.3257 | 0.2594 | 0.3447 | 0.2772 | 0.3669 |
| | **%Promotion** | **5.9%** | **6.0%** | **20.3%** | **15.5%** | **20.1%** | **14.8%** | **33.0%** | **20.8%** | **30.7%** | **18.3%** | **35.3%** | **20.0%** | **23.7%** | **15.5%** |
| Elec. | Vanilla | 0.1753 | 0.2904 | 0.1714 | 0.2895 | 0.6550 | 0.6305 | 0.5761 | 0.5749 | 0.6783 | 0.6352 | 0.2733 | 0.3632 | 0.4216 | 0.4639 |
| | with TATO | 0.1729 | 0.2859 | 0.1710 | 0.2885 | 0.5539 | 0.5566 | 0.4434 | 0.4949 | 0.5100 | 0.5350 | 0.2716 | 0.3629 | 0.3538 | 0.4206 |
| | **%Promotion** | **1.4%** | **1.6%** | **0.3%** | **0.3%** | **15.4%** | **11.7%** | **23.0%** | **13.9%** | **24.8%** | **15.8%** | **0.6%** | **0.1%** | **16.1%** | **9.3%** |
| Exch. | Vanilla | 0.5190 | 0.5004 | 0.8317 | 0.7013 | 0.2897 | 0.3855 | 0.2524 | 0.3615 | 0.2450 | 0.3570 | 0.4013 | 0.4404 | 0.4232 | 0.4577 |
| | with TATO | 0.3073 | 0.3905 | 0.2881 | 0.3727 | 0.2797 | 0.3801 | 0.2570 | 0.3586 | 0.2635 | 0.3673 | 0.2529 | 0.3541 | 0.2748 | 0.3706 |
| | **%Promotion** | **40.8%** | **22.0%** | **65.4%** | **46.9%** | **3.5%** | **1.4%** | -1.8% | **0.8%** | -7.5% | -2.9% | **37.0%** | **19.6%** | **35.1%** | **19.0%** |
| Traffic | Vanilla | 0.0605 | 0.1380 | 0.0616 | 0.1404 | 0.3646 | 0.4982 | 0.3364 | 0.4690 | 0.3049 | 0.4233 | 0.0762 | 0.1548 | 0.2007 | 0.3040 |
| | with TATO | 0.0589 | 0.1353 | 0.0602 | 0.1384 | 0.3326 | 0.4663 | 0.2743 | 0.4112 | 0.2519 | 0.3766 | 0.0769 | 0.1554 | 0.1758 | 0.2805 |
| | **%Promotion** | **2.5%** | **2.0%** | **2.3%** | **1.5%** | **8.8%** | **6.4%** | **18.5%** | **12.3%** | **17.4%** | **11.0%** | -0.9% | -0.4% | **12.4%** | **7.7%** |
| Weather | Vanilla | 0.4937 | 0.4967 | 0.5888 | 0.5533 | 0.4984 | 0.4785 | 0.4929 | 0.4606 | 0.5422 | 0.4958 | 0.7282 | 0.5692 | 0.5573 | 0.5090 |
| | with TATO | 0.6048 | 0.5290 | 0.5898 | 0.5343 | 0.4944 | 0.4814 | 0.4961 | 0.4687 | 0.5107 | 0.4818 | 0.5980 | 0.5245 | 0.5490 | 0.5033 |
| | **%Promotion** | -22.5% | -6.5% | -0.2% | **3.4%** | **0.8%** | -0.6% | -0.7% | -1.8% | **5.8%** | **2.8%** | **17.9%** | **7.9%** | **1.5%** | **1.1%** |
| **Average** | Vanilla | 0.3431 | 0.4090 | 0.3923 | 0.4419 | 0.4185 | 0.4804 | 0.4066 | 0.4661 | 0.4031 | 0.4594 | 0.3770 | 0.4170 | 0.3901 | 0.4456 |
| | with TATO | 0.3225 | 0.3920 | 0.2950 | 0.3723 | 0.3799 | 0.4513 | 0.3568 | 0.4323 | 0.3465 | 0.4228 | 0.3225 | 0.3877 | 0.3372 | 0.4097 |
| | **%Promotion** | **6.0%** | **4.1%** | **24.8%** | **15.7%** | **9.2%** | **6.0%** | **12.2%** | **7.3%** | **14.0%** | **8.0%** | **14.5%** | **7.0%** | **13.6%** | **8.1%** |

Table 1 demonstrates that TATO commonly matches or outperforms vanilla baselines in 84.3% of the 96 evaluated cases. The MSE improvements range from 6.0% to 24.8% across different LTMs, with an average reduction of 13.6%. Across all LTMs in each dataset, TATO consistently outperforms the vanilla baselines, with the lowest improvement of 1.5% on Weather and the greatest of 35.1% on Exchange. Notably, TATO delivers substantial gains precisely in cases where vanilla baselines perform poorly (highlighted in blue), validating our motivation that ineffective predictions can be effectively mitigated through adaptive transformation optimization. In a few cases, such as Timer-UTSD on Weather, Moirai-large on Exchange, and Chronos-tiny on ETTh2, TATO exhibits limited gains, suggesting that even with adaptive transformation optimization, distribution shifts can still pose challenges. We leave further investigation of these edge cases to future work.

Besides, we present the distribution shifts of MAE in three representative tasks caused by TATO in Figure 3. The MAE distribution of vanilla, shown in the yellow bars, shifts to TATO's blue bars over a smaller, narrower range, indicating a significant performance improvement.

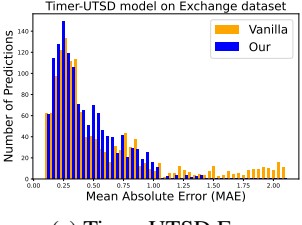
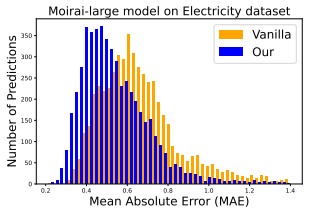
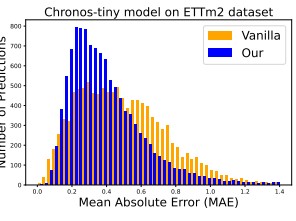

(a) Timer-UTSD Error.      (b) Moirai-large Error.      (c) Chronos-tiny Error.

Figure 3: Distribution of MAE before and after applying TATO on three representative tasks. Across all datasets, TATO consistently shifts the error distribution toward lower values, indicating improved forecasting accuracy compared to the vanilla baseline.

## 4.4 EFFICIENCY

To evaluate efficiency, we calculate the overhead introduced by TATO during the optimization process. Table 2 presents the overhead data for three representative models: Timer-UTSD, Moirai-base, and Chronos-tiny, with fixed prediction lengths of 96. The experiments were conducted under nearly identical settings, except for the number of trials and samples. The number of trials controls the optimization budget, and the number of samples may vary according to the data conditions in practice.

Table 2: Overhead in TATO optimization phase across different models and configurations. Even when configured with 500 trials or 500 samples, the time overhead remains reasonably acceptable.

| Model | **Trials** | | | **Samples** | | |
|---|---|---|---|---|---|---|
|  | 25 | 100 | 500 | 25 | 100 | 500 |
| Timer-UTSD | 2.97 s | 11.21 s | 76.56 s | 6.80 s | 11.21 s | 37.44 s |
| Moirai-base | 5.61 s | 18.05 s | 106.40 s | 11.39 s | 18.05 s | 65.90 s |
| Chronos-tiny | 29.05 s | 112.39 s | 573.30 s | 87.71 s | 112.39 s | 398.03 s |

Three different numbers of trials are used in the first part of Table 2 to illustrate the changes in overhead over trials, with the sample size fixed at 100. Similarly, three different sample sizes are used in the second part of Table 2 to illustrate the changes in overhead across samples, with the trials fixed at 100. During the optimization phase, the primary source of overhead is the additional model inference required by our method to evaluate the effectiveness of data processing, which is heavily influenced by the model's inference time. Most experiments can be completed within 120 seconds, which is highly efficient compared to finetuning or test-time adaptation methods that require additional training. Chronos takes longer because it relies on LLMs. In general, the optimization overhead scales linearly with the number of optimization trials and samples.

In the inference phase, we assess the overhead of transformations for different batch sizes. During the inference phase, extra overhead comes from the data transformations selected by TATO. When the batch size is 1, the additional overhead is under 3 milliseconds, indicating that the tailored transformation operators are highly efficient relative to model inference. The extra overhead varies slightly across models because different transformations are selected in different scenarios.

## 4.5 ENHANCING UNIVERSALLY FINETUNED LTMS

In Section 2.1, we compare TATO with finetuning in terms of the trade-off between specificity and generalization. One way to preserve a single LTM's generalizability of finetuning is to train jointly across all downstream domains. In this experiment, we apply TATO to such universally finetuned models to further demonstrate its effectiveness. We finetune Timer-UTSD and Timer-LOTSA separately on all eight datasets, using 500 samples per dataset, to obtain a single LTM

that generalizes well across domains. We then use TATO to optimize transformation pipelines for each finetuned model. As shown in Table 3, TATO further improves forecasting performance, on average, beyond the finetuned baselines. For Timer-UTSD finetuned on all datasets, TATO achieves a 4.5% reduction in MSE and a 4.1% reduction in MAE. For Timer-LOTSA finetuned on all datasets, TATO achieves a 9.9% reduction in MSE and a 6.5% reduction in MAE. On average, TATO delivers a 7.3% MSE improvement and a 5.3% MAE improvement. These results demonstrate that TATO and finetuning address complementary aspects of domain adaptation: while finetuning adjusts model parameters to capture cross-domain patterns, TATO further enhances specificity by optimizing data transformations for each domain, together enabling more effective FrozenForecasting.

Table 3: Results of prediction using TATO upon finetuning. The average results under prediction lengths of {24,48,96,192} are reported. Positive %Promotion indicates performance enhancement.

| | ETTh1 | | ETTh2 | | ETTm1 | | ETTm2 | | Elec. | | Exch. | | Traffic | | Weather | | Average | |
|---|---|---|---|---|---|---|---|---|---|---|---|---|---|---|---|---|---|---|---|
| | MSE | MAE | MSE | MAE | MSE | MAE | MSE | MAE | MSE | MAE | MSE | MAE | MSE | MAE | MSE | MAE | MSE | MAE |
| **Timer-UTSD** | 0.3821 | 0.4692 | 0.3016 | 0.4311 | 0.2434 | 0.3692 | 0.2746 | 0.4053 | 0.2576 | 0.3604 | 0.4382 | 0.4737 | 0.0922 | 0.1951 | 0.5052 | 0.5106 | 0.3119 | 0.4018 |
| with TATO | 0.3765 | 0.4645 | 0.2992 | 0.4273 | 0.2446 | 0.3677 | 0.2624 | 0.3757 | 0.2421 | 0.3476 | 0.3144 | 0.3908 | 0.0899 | 0.1905 | 0.5523 | 0.5188 | 0.2976 | 0.3853 |
| **%Promotion** | **1.5%** | **1.0%** | **0.8%** | **0.9%** | **-0.5%** | **0.4%** | **4.4%** | **7.3%** | **6.0%** | **3.6%** | **28.3%** | **17.5%** | **2.5%** | **2.4%** | **-9.3%** | **-1.6%** | **4.5%** | **4.1%** |
| **Timer-LOTSA** | 0.3867 | 0.4705 | 0.2949 | 0.4235 | 0.2540 | 0.3730 | 0.2721 | 0.3984 | 0.2677 | 0.3672 | 0.4690 | 0.4966 | 0.1001 | 0.2133 | 0.4881 | 0.4937 | 0.3166 | 0.4045 |
| with TATO | 0.3937 | 0.4751 | 0.3000 | 0.4281 | 0.2317 | 0.3501 | 0.2309 | 0.3421 | 0.2618 | 0.3671 | 0.2928 | 0.3789 | 0.0928 | 0.2019 | 0.4757 | 0.4807 | 0.2849 | 0.3780 |
| **%Promotion** | **-1.8%** | **-1.0%** | **-1.7%** | **-1.1%** | **8.8%** | **6.1%** | **15.2%** | **14.1%** | **2.2%** | **0.03%** | **37.6%** | **23.7%** | **7.3%** | **5.4%** | **2.5%** | **2.6%** | **9.9%** | **6.5%** |
| **Average** | 0.3844 | 0.4699 | 0.2983 | 0.4273 | 0.2487 | 0.3711 | 0.2734 | 0.4012 | 0.2627 | 0.3638 | 0.4536 | 0.4852 | 0.0962 | 0.2042 | 0.4967 | 0.5022 | 0.3142 | 0.4032 |
| with TATO | 0.3851 | 0.4698 | 0.2996 | 0.4277 | 0.2382 | 0.3589 | 0.2467 | 0.3589 | 0.2519 | 0.3573 | 0.3036 | 0.3848 | 0.0913 | 0.1962 | 0.5140 | 0.4997 | 0.2913 | 0.3816 |
| **%Promotion** | **-0.2%** | **0.01%** | **-0.4%** | **-0.1%** | **4.2%** | **3.3%** | **9.8%** | **10.7%** | **4.1%** | **1.8%** | **33.1%** | **20.7%** | **5.0%** | **3.9%** | **-3.5%** | **0.5%** | **7.3%** | **5.3%** |

## 4.6 ANALYSIS EXPERIMENTS

The analysis experiments were conducted on three representative LTMs, as shown in Table 2, with a fixed prediction length of 96. We first examine TATO's scalability by varying the number of transformation trials and the number of data samples. The impact of specific transformations or pipeline steps is also analyzed.

### 4.6.1 SCALABILITY

We conducted scalability experiments to determine the minimum required sample and trial sizes to achieve significant performance gains, and to explore the upper bounds on improvement achievable through data transformations. Figure 4a shows the effect of increasing the number of transformation trials on MSE reduction (%Promotion), with a fixed sample size of 100. Results show a clear positive trend, with MSE improvement increasing from 7.2% with 50 trials to 9.4% with 500 trials, indicating that a more extensive search consistently yields better transformations. Figure 4b examines the impact of sample size on optimization effectiveness, with a fixed budget of 100 trials. As more samples become available, both the mean improvement and its stability increase, with the MSE reduction increasing from 4.6% with 50 samples to 8.8% with 500 samples, while the variance notably decreases. These findings confirm that TATO benefits from both more trials and more data, with practical resource requirements well within affordable ranges.

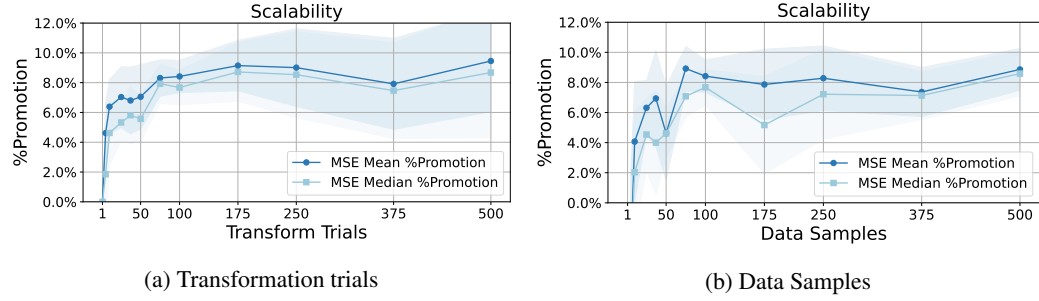

(a) Transformation trials

(b) Data Samples

Figure 4: Scalability analysis of TATO. (a) MSE improvement with increasing transformation trials (fixed 100 samples). (b) MSE improvement with increasing sample size (fixed 100 trials). Performance consistently improves with more trials and data, ranging from 50 to 500 in both dimensions.

### 4.6.2 ABLATIONS

We conducted ablation studies to identify the effects of transformations and steps in TATO's framework. Figure 5 presents ablation studies evaluating the contribution of each transformation operator and framework component to TATO's overall performance. The full configuration, incorporating all operators (Section A.3) and steps (Section 3.2), achieves the best mean and median MSE reduction, confirming that each element contributes positively to forecasting accuracy and stability. Notably, removing the Trimmer or Scaler operators results in a substantial performance drop, underscoring their critical role in preparing data for frozen LTMs. Interestingly, omitting the Denoiser operator or the TwoStageRank step yields greater mean improvement but at the cost of increased variance and lower median performance. This suggests that while some components may be particularly beneficial for specific tasks, they also help ensure robustness across diverse scenarios. Overall, the results validate the necessity of TATO's complete design for achieving consistent and reliable gains.

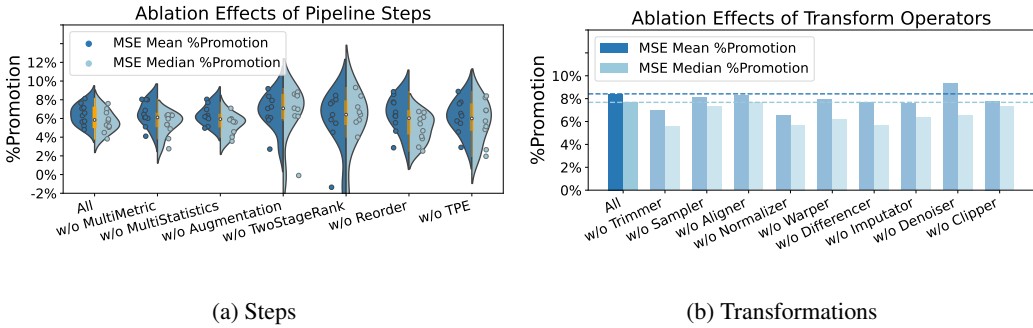

(a) Steps            (b) Transformations

Figure 5: Ablation study results. (a) Effect of removing key framework components on the reduction of MSE. (b) Effect of removing individual transformation operators on MSE reduction. Mean and median %Promotion are shown for each variant.

The ablation results reveal an interesting trade-off between mean performance and robustness. While removing Denoiser reduces the average MSE, the increased variance and lower median suggest that it plays a crucial role in stabilizing predictions across diverse samples. Similarly, the TwoStageRank mechanism, despite slightly reducing mean gains, proves essential for consistent performance by eliminating trials that excel only on specific metrics. This validates our design philosophy: robustness across domains is as important as peak performance in FrozenForecasting.

## 5 CONCLUSION

In this paper, we introduced TATO, an automated framework that optimizes transformation pipelines to enable effective FrozenForecasting with pretrained LTMs. Through nine tunable operators addressing context, scale, and outliers, combined with a two-stage Pareto-based ranking mechanism, TATO consistently improves forecasting performance across diverse domains. Extensive experiments demonstrate an average MSE reduction of 13.6% and a maximum of 65.4%, with optimization typically under two minutes. These results validate TATO as a practical, lightweight solution for unlocking the full potential of frozen LTMs without costly finetuning. Future work includes extending TATO to multivariate settings, expanding the set of operators for specialized domains, exploring adaptive operator selection, and integrating test-time adaptation to enhance robustness to distribution shifts. We believe this data-centric perspective opens promising avenues for more sustainable and generalizable deployment of large models across real-world applications.

### ACKNOWLEDGMENTS

This work is supported by the National Key Research and Development Program of China (2023YFB3308003), the BNRist Project, and the National Engineering Research Center for Big Data Software. We also gratefully acknowledge Mengren Zheng's contributions to the additional experiments during rebuttal.

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

# A APPENDIX

## A.1 THE DETAILS OF USAGE OF LLMs

We use the Pro version of Grammarly to check spelling and polish complex sentences. LLMs are probably used inside Grammarly. We did not use LLMs in any other way in this paper.

## A.2 IMPLEMENTATION DETAILS

All experiments were conducted on a server equipped with Intel(R) Xeon 14-core processors, 384GB of RAM, and 8 NVIDIA TITAN X (Pascal) GPUs, running Ubuntu 18.04.

### A.2.1 Dataset Details

We evaluate the models on a diverse set of time series datasets from real scenarios, including (1) ETT(Zhou et al., 2021) (Electricity Transformer Temperature), comprising hourly data from ETTh1 and ETTh2 and 15-minute data from ETTm1 and ETTm2, collected between July 2016 and July 2018, including load and oil temperature. (2) Electricity[1], containing hourly electricity consumption data of 321 customers from 2012 to 2014. (3) Exchange [2], recording daily exchange rates of eight countries from 1990 to 2016. (4) Traffic[3], a collection of hourly road occupancy rates measured by sensors on San Francisco Bay area freeways. (5) Weather[4], containing meteorological data recorded every 10 minutes throughout 2020, including 21 indicators like air temperature and humidity.

When possible, we follow the same data preparation and train-validation-test set split protocol used in TimesNet(Wu et al., 2022). TATO optimizes the transformation pipelines on frozen LTMs using only a limited subset (e.g., 100 or 500 samples) of the training set. The data splitting is consistent with TimesNet to ensure a fair comparison. For instance, ETTm1 and ETTm2 have datasets split (34465, 11521, 11521) for training, validation, and testing. Only less than 2% of the total training samples are used in TATO, and the test set is consistently kept unseen during the optimization of the transformation pipeline.

In time series forecasting, short-term forecasting tasks usually use $\{6,12,24,48\}$ as prediction lengths, while long-term forecasting tasks usually use $\{96,192,336,720\}$. We wanted to cover both scenarios in a reasonable experiment workload, so $\{24,48,96,192\}$ is picked. For the forecasting settings, we fix the prediction length to vary in $\{24,48,96,192\}$. Our univariate prediction experiments focus exclusively on each dataset's OT column, representing the most meaningful data for real-world production scenarios.

### A.2.2 Large Time Series Models

Several advanced transformer-based LTMs designed for univariate time series forecasting have been proposed recently. Timer(Liu et al., 2024b) employs a GPT-style architecture and benefits from extensive pre-training, enabling it to autoregressively predict the next time series token in a unified generative manner. Timer-UTSD and Timer-LOTSA maintain the same training configuration but are trained on different datasets: the Unified Time Series Dataset (UTSD) and the Large-scale Open Time Series Archive (LOTSA), respectively. Moirai(Woo et al., 2024), a masked encoder-based universal time series forecasting transformer trained on LOTSA, demonstrates superior performance as a zero-shot forecaster compared to full-shot models. Chronos(Fatir Ansari et al., 2024) tokenizes time series values using scaling and quantization into a fixed vocabulary and trains existing transformer-based language model architectures on these tokenized time series using cross-entropy loss. The inference parameters of the LTMs were fixed. Timer followed the settings in the original paper, with patch_size of 96. Moirai used a patch_size of 128 and num_samples of 20. Chronos used a patch_len of 512 and num_samples of 3.

### A.2.3 Hyperparameter Space

The hyperparameters in the table are designed to optimize various data transformation operations for time series forecasting models. For example, the *trimmer_length* is a discrete hyperparameter that controls how much past data is retained, with values ranging from 5 to 14 times 96 and an original setting of 7 times 96. The *sampler_factor* is a continuous hyperparameter that adjusts the downsampling or upsampling rate, with a range of 1 to 2, starting with a value of 1, meaning no resampling initially. The *aligner* is a categorical hyperparameter that determines the padding method, offering options like 'none', 'edge', and *scaler* is used to select the normalization technique, with choices like 'none', 'standard', and 'robust', also defaulting to 'none'. The *differencer_level* is a discrete hyperparameter that specifies the order of differencing, either 0 or 1, with 0 indicating no differencing. The *warper* governs the application of warping techniques, such as the 'log' transformation, with 'none' as the default setting. The *outlier_detect* determines how outliers are identified, offering options like 'none', '3_sigma', and '1.5_iqr', with 'none' as the starting point. The *denoiser*

---

[1]https://archive.ics.uci.edu/ml/datasets/ElectricityLoadDiagrams20112014

[2]http://pems.dot.ca.gov

[3]http://pems.dot.ca.gov

[4]https://www.bgc-jena.mpg.de/wetter/

specifies the denoising technique, such as 'none' or 'ewma', with the default being 'none'. Finally, the *clipper_factor* sets the range for clipping extreme values, with options like 'none', '0.5', and '1', starting with 'none' unless clipping is needed. These hyperparameters are tailored to enhance model performance by refining the input data through various preprocessing techniques.

Table 4: Hyperparameters for Data transform

| Hyperparameter | Type | Value Range |
|---|---|---|
| trimmer_length | Discrete | [5*96, 6*96, ..., 14*96] |
| sampler_factor | Continuous | [1, 2] |
| aligner | Categorical | {'none', 'edge', 'mean'} |
| scaler | Discrete | {'none', 'standard', 'robust'} |
| differencer_level | Discrete | [0, 1] |
| warper | Categorical | {'none', 'log'} |
| outlier_detect | Categorical | {'none', '3_sigma', '1.5_iqr'} |
| denoiser | Categorical | {'none', 'ewma'} |
| clipper_factor | Categorical | {'none', '0.5', '1'} |

### A.3 TIME SERIES TRANSFORMATIONS

The recent LTMs generally adopt several fixed data transformations. They generally embed time series points using patching techniques. Some LTMs segment data into multiple scales to extract more features and employ stochastic sampling to generate robust forecasting distributions. The original transformations of the LTMs are typically carefully designed in accordance with the pretraining datasets. However, the pretraining datasets may differ from the test data, making LTMs sensitive to the transformations. The proposed framework, TATO, enhances the reliability of LTMs by employing time-series transformation optimizations rather than a fixed sequence of transformations. TATO's candidate transformations involve nine operators of three types. TATO can be easily extended to other transformations. However, a vast search space poses a significant challenge. We carefully selected the nine transformations to cover the three most representative dimensions of transformation, and they exhibit a strong capability of enhancing LTMs in experiments.

#### A.3.1 CONTEXTUAL TRANSFORMATIONS

The first is the use of contextual transformations to slice inference inputs, such as adjusting the lookback length, downsampling, and patch completion. The operator Trimmer adjusts the lookback length to approach the fit range for the forecasting task (Lee et al., 2021). A longer lookback means more information from recent history. Generally, more information is better. However, too much information could interfere with LTMs, resulting in poor performance. We denote Trimmer as follows:

$$T_{\text{trimmer}}(\mathbf{x}, \mathbf{P_l}) = \mathbf{x}[\text{len}(\mathbf{x}) - \mathbf{P_l} : \text{len}(\mathbf{x})] \tag{1}$$

where $\mathbf{x}$ is a lookback sample whose length is denoted by $\text{len}(\mathbf{x})$, and $\mathbf{P_l}$ represents the hyperparamter of lookback length.

The operator Sampler modifies the frequency of data points to find an optimal balance between details and overall trend (Liu et al., 2022). A choice of Sampler, *Downsampling*, can reduce noise and highlight trends and cycles, whereas another choice *Upsampling* intends to provide more detailed data. We denote Downsampling as follows:

$$T_{\text{sampler(Downsampling)}}(\mathbf{x}, \mathbf{P_s}) = \mathbf{x}[:: \mathbf{P_s}] \tag{2}$$

where $\mathbf{x}[:: \mathbf{P_s}]$ means sampling at every $\mathbf{P_s}$ points and $\mathbf{P_s}$ is a hyperparameter to optimize.

The operator Aligner addresses the issue of patch alignment for LTMs. The recent LTMs typically divide sequences of data points into patches for autoregressive forecasting. If the input data length is not an integral multiple of the patch size, LTMs may utilize padding with zeros, which introduces low-quality data. Aligner provides several typical methods of padding the sequences, which may increase the quality of input data. We denote one choice of Aligner, *Mean*, as follows:

$$T_{\text{aligner(Mean)}}(\mathbf{x}) = \text{pad}(\mathbf{x}, \text{mean}(\mathbf{x})) \tag{3}$$

where $\text{pad}(\mathbf{x}, v)$ is a function that appends $v$ to $\mathbf{x}$ to make the length reach an integral multiple of the patch size.

### A.3.2 NORMALIZING TRANSFORMATIONS

The second is normalizing transformations, which adjust the samples' values to a constraint range. Normalizing transformations ensures a consistent distribution of input samples' values to help LTMs perform stably(Bhanja & Das, 2018). The common normalization methods include the standard scaler, which scales data to have zero mean and unit variance, and the robust scaler, which is less sensitive to outliers. We denote a choice of Scaler, *STD*, as follows:

$$T_{\text{scaler(STD)}}(\mathbf{x}) = \frac{\mathbf{x} - \mu}{\sigma} \tag{4}$$

where $\mu$ and $\sigma$ represent mean and variance of $\mathbf{x}$ seperately.

The operator Differencer transforms the data by computing the differences between adjacent points(Kwiatkowski et al., 1992). It effectively removes acute trends and reduces the variability in mean and variance over time, making the data more stationary. Such transformations are crucial for time series models that rely on linear or stationary assumptions. We denote the first-order choice of Differencer as follows:

$$T_{\text{differencer(1)}}(\mathbf{x}) = \{\mathbf{x}_t - \mathbf{x}_{t-1} | t = 1, ..., \text{len}(\mathbf{x})\} \tag{5}$$

The operator Warper applies transformations, such as logarithmic adjustments, to stabilize the variance(Box & Cox, 1964). This approach is particularly beneficial for data with skewed distributions or extreme peaks. We denote a choice of Warper, *LOG*, as follows:

$$T_{\text{warper(LOG)}}(\mathbf{x}) = \log(1 + |\mathbf{x}|) \cdot \text{sign}(\mathbf{x}) \tag{6}$$

### A.3.3 OUTLIER TRANSFORMATIONS

The third is outlier transformations, which resolve outliers or noise to ensure a stable input for LTMs. The operator Denoiser aims to reduce noise in the data, making trends and seasonal patterns more prominent (Brown, 1959). Techniques such as *weighted moving average* and fast Fourier transform filtering can smooth the data. By increasing the signal-to-noise ratio, these methods may improve LTMs' ability to make more accurate predictions. We denote a choice of Denoiser, *MA*, as follows:

$$T_{\text{denoisor(MA)}}(\mathbf{x}, \mathbf{P}_\omega) = \frac{1}{\mathbf{P}_\omega} \sum_{i=1}^{\mathbf{P}_\omega} \mathbf{x}_{t-i} | t = \mathbf{P}_\omega, ..., \text{len}(\mathbf{x}) \tag{7}$$

where $\mathbf{P}_\omega$ is a hyperparameter for the window size of the moving average algorithm.

The Imputator operator replaces anomalies with normal values using methods such as linear interpolation. Outliers can be detected using methods such as k-sigma(Blázquez-García et al., 2021) and k-IQR(Saradjian & Akhoondzadeh, 2011). We denote a choice of Imputator, *IQR*, as follows:

$$T_{\text{imputator(IQR)}}(\mathbf{x}, \mathbf{P_k}) = \begin{cases} \mathbf{x}_t & |\mathbf{x}_t - x_M| \le \mathbf{P_k} \cdot \text{IQR}(\mathbf{x}) \\ \frac{1}{2}(\mathbf{x}_{t-1} + \mathbf{x}_{t+1}) & \text{otherwise} \end{cases} \tag{8}$$

where $\mathbf{P_k}$ is a hyperparameter for severity adjustment.

The Clipper operator addresses abnormal data shifts by utilizing the IQR bounds of the original data to clip the extensive inverse-transformed data. It ensures that extreme values are within a reasonable range, preventing unrealistic predictions. We denote Clipper as follows:

$$T_{\text{clipper}}(\mathbf{x}, \mathbf{P_k}) = \max(\min(\mathbf{x}, \text{Q3} + \text{IQR} \cdot \mathbf{P_k}), \text{Q1} - \text{IQR} \cdot \mathbf{P_k}) \tag{9}$$

where Q1 and Q3 represent the first and third quartiles, and $\mathbf{P_k}$ is a hyperparameter for adjustment of clipping range.

The preceding section describes the nine operators currently adopted in the TATO framework. As LTMs and transformation methods advance, the TATO framework will also evolve, with LTMs performing more diverse tasks and the transformation pipeline comprising a greater number of operators.

### A.4 DETAILS OF THE PARETO-BASED RANKING OF PERFORMANCE

A key challenge in evaluating transformation pipelines is that no single trial consistently outperforms others across all metrics; averaging all metrics risks giving poorly performing trials a favorable ranking. To address this, we adopt a two-stage Pareto-based ranking approach (Palakonda & Mallipeddi, 2017) to select the final transformation pipeline.

In the first stage, we filter out underperforming trials using metric subsets computed on all augmented samples. Specifically, we construct multiple metric combinations (e.g., MSE, MAE, MAPE and MSE, MSPE) and retain only those trials that rank among the best on all combinations, forming a Pareto set. This eliminates trials that excel only on specific metrics while performing poorly on others. In practice, we set the filtering threshold to exclude the worst-performing trials, balancing risk reduction with computational efficiency.

In the second stage, we rank the remaining trials using only the original (non-augmented) samples. The final ranking is computed as a weighted sum of all metric ranks, with MSE assigned the highest weight (3), followed by MAE (2), and other error metrics each receiving a weight of 1. This weighting scheme reflects mainstream practice in time series forecasting while remaining adjustable to downstream task requirements. The top-ranked transformation pipeline is then applied to test data for FrozenForecasting.

### A.5 DETAILED RESULTS

Table 5 provides detailed results of MSE and MAE for predictions across various models and tasks using TATO. The prediction lengths range from 24 to 192, covering both shorter and longer forecasting horizons. Generally, the error metrics tend to increase with longer prediction lengths, likely due to the increased uncertainty and complexity in capturing long-term dependencies in the data.

Table 5: Prediction results using TATO with various models and datasets.

| Models | | Timer-UTSD | | Timer-LOTSA | | Moirai-small | | Moirai-base | | Moirai-large | | Chronos-tiny | |
|---|---|---|---|---|---|---|---|---|---|---|---|---|---|
| Error Metric | | MSE | MAE | MSE | MAE | MSE | MAE | MSE | MAE | MSE | MAE | MSE | MAE |
| ETTh1 | 24 | 0.2521 | 0.3850 | 0.2485 | 0.3724 | 0.2615 | 0.3922 | 0.2617 | 0.3893 | 0.2431 | 0.3724 | 0.2716 | 0.3930 |
| | 48 | 0.3469 | 0.4511 | 0.4671 | 0.5289 | 0.3707 | 0.4659 | 0.3783 | 0.4658 | 0.3526 | 0.4488 | 0.3922 | 0.4788 |
| | 96 | 0.4567 | 0.5177 | 0.4782 | 0.5254 | 0.4801 | 0.5274 | 0.5305 | 0.5620 | 0.4650 | 0.5173 | 0.4989 | 0.5353 |
| | 192 | 0.5048 | 0.5509 | 0.5578 | 0.5773 | 0.6715 | 0.6322 | 0.6306 | 0.6155 | 0.5956 | 0.5910 | 0.6214 | 0.6075 |
| ETTh2 | 24 | 0.2121 | 0.3505 | 0.2187 | 0.3467 | 0.3803 | 0.4767 | 0.3320 | 0.4349 | 0.2683 | 0.3937 | 0.2446 | 0.3640 |
| | 48 | 0.2822 | 0.4113 | 0.3285 | 0.4322 | 0.4225 | 0.5057 | 0.3927 | 0.4820 | 0.3230 | 0.4379 | 0.3483 | 0.4482 |
| | 96 | 0.3729 | 0.4762 | 0.3979 | 0.5022 | 0.4874 | 0.5466 | 0.4600 | 0.5315 | 0.3921 | 0.4887 | 0.4724 | 0.5323 |
| | 192 | 0.4140 | 0.5120 | 0.4274 | 0.5328 | 0.5862 | 0.6086 | 0.6303 | 0.6361 | 0.4920 | 0.5564 | 0.6536 | 0.6559 |
| ETTm1 | 24 | 0.1179 | 0.2501 | 0.0884 | 0.2114 | 0.1038 | 0.2294 | 0.1014 | 0.2276 | 0.0989 | 0.2253 | 0.0993 | 0.2239 |
| | 48 | 0.2386 | 0.3654 | 0.1538 | 0.2912 | 0.1619 | 0.3021 | 0.1566 | 0.2943 | 0.1576 | 0.2947 | 0.1823 | 0.3153 |
| | 96 | 0.3399 | 0.4374 | 0.2430 | 0.3716 | 0.2664 | 0.3909 | 0.2739 | 0.3949 | 0.2626 | 0.3860 | 0.2805 | 0.3965 |
| | 192 | 0.4595 | 0.5110 | 0.3903 | 0.4779 | 0.3713 | 0.4645 | 0.3824 | 0.4744 | 0.3821 | 0.4719 | 0.4207 | 0.4888 |
| ETTm2 | 24 | 0.1968 | 0.2861 | 0.1049 | 0.2068 | 0.1120 | 0.1973 | 0.1052 | 0.1955 | 0.0838 | 0.1791 | 0.0925 | 0.1795 |
| | 48 | 0.4643 | 0.5114 | 0.2150 | 0.3246 | 0.2379 | 0.3297 | 0.2520 | 0.3407 | 0.2467 | 0.3215 | 0.2251 | 0.3219 |
| | 96 | 0.5246 | 0.5652 | 0.3043 | 0.4125 | 0.2639 | 0.3891 | 0.2949 | 0.4014 | 0.2476 | 0.3665 | 0.2964 | 0.3937 |
| | 192 | 0.5617 | 0.6008 | 0.3784 | 0.4642 | 0.3357 | 0.4457 | 0.3526 | 0.4532 | 0.3329 | 0.4356 | 0.4236 | 0.4840 |
| Electricity | 24 | 0.1219 | 0.2442 | 0.1208 | 0.2451 | 0.4794 | 0.5175 | 0.4256 | 0.4826 | 0.4355 | 0.4922 | 0.1927 | 0.3060 |
| | 48 | 0.1560 | 0.2741 | 0.1537 | 0.2762 | 0.5375 | 0.5507 | 0.4449 | 0.4954 | 0.5173 | 0.5392 | 0.2471 | 0.3470 |
| | 96 | 0.1913 | 0.3014 | 0.1867 | 0.3039 | 0.6047 | 0.5816 | 0.4415 | 0.4970 | 0.5434 | 0.5549 | 0.2985 | 0.3821 |
| | 192 | 0.2224 | 0.3237 | 0.2227 | 0.3289 | 0.5940 | 0.5765 | 0.4615 | 0.5047 | 0.5438 | 0.5537 | 0.3481 | 0.4165 |
| Exchange | 24 | 0.0784 | 0.2134 | 0.0732 | 0.2013 | 0.0825 | 0.2139 | 0.0775 | 0.2086 | 0.0720 | 0.1976 | 0.0705 | 0.1962 |
| | 48 | 0.1509 | 0.2901 | 0.1450 | 0.2849 | 0.1676 | 0.3137 | 0.1502 | 0.2870 | 0.1640 | 0.3025 | 0.1383 | 0.2752 |
| | 96 | 0.3634 | 0.4396 | 0.3027 | 0.4098 | 0.3160 | 0.4109 | 0.2909 | 0.3986 | 0.3099 | 0.4087 | 0.2929 | 0.3962 |
| | 192 | 0.6367 | 0.6191 | 0.6314 | 0.5948 | 0.5525 | 0.5821 | 0.5094 | 0.5401 | 0.5080 | 0.5604 | 0.5099 | 0.5488 |
| Traffic | 24 | 0.0557 | 0.1307 | 0.0570 | 0.1335 | 0.3166 | 0.4608 | 0.2860 | 0.4302 | 0.2427 | 0.3675 | 0.0644 | 0.1370 |
| | 48 | 0.0586 | 0.1338 | 0.0597 | 0.1371 | 0.3317 | 0.4656 | 0.2806 | 0.4136 | 0.2640 | 0.3858 | 0.0739 | 0.1490 |
| | 96 | 0.0609 | 0.1371 | 0.0621 | 0.1399 | 0.3574 | 0.4786 | 0.2638 | 0.3983 | 0.2592 | 0.3800 | 0.0798 | 0.1601 |
| | 192 | 0.0606 | 0.1394 | 0.0621 | 0.1430 | 0.3249 | 0.4602 | 0.2670 | 0.4027 | 0.2419 | 0.3733 | 0.0893 | 0.1753 |
| Weather | 24 | 0.2650 | 0.3321 | 0.2227 | 0.2954 | 0.2207 | 0.2981 | 0.2259 | 0.2984 | 0.2428 | 0.3036 | 0.2250 | 0.3033 |
| | 48 | 0.5237 | 0.4856 | 0.6100 | 0.5638 | 0.3969 | 0.4306 | 0.3841 | 0.4140 | 0.3986 | 0.4281 | 0.4225 | 0.4456 |
| | 96 | 0.9170 | 0.6769 | 0.6858 | 0.5922 | 0.6100 | 0.5666 | 0.5982 | 0.5367 | 0.6146 | 0.5540 | 0.8986 | 0.6746 |
| | 192 | 0.7136 | 0.6214 | 0.8408 | 0.6857 | 0.7501 | 0.6302 | 0.7764 | 0.6259 | 0.7870 | 0.6415 | 0.8459 | 0.6745 |

## A.6 ON DIFFERENT PERSPECTIVES

Table 6 provides additional results from Section 4.3. We analyze the statistics of MSE and MAE, including Mean, Median, STD, and IQR, to calculate %Promotions across various scenarios. From a model perspective, TATO consistently enhances performance across all models. In particular, Moirai shows that our method improves forecasting performance more effectively with larger models.

Table 6: Results of prediction using TATO from different perspectives. Percentage improvements (% Promotion) are grouped and averaged by model, data, and task.

| %Promotion of Error Metric Statistics | | MSE Mean %Promotion | MAE Mean %Promotion | MSE Median %Promotion | MAE Median %Promotion | MSE STD %Promotion | MAE STD %Promotion | MSE IQR %Promotion | MAE IQR %Promotion |
|---|---|---|---|---|---|---|---|---|---|
| **Model** | Timer-UTSD | 6.00% | 4.10% | -1.40% | 0.90% | 6.80% | 8.50% | 5.10% | 6.00% |
| | Timer-LOTSA | 24.80% | 15.70% | 17.10% | 13.40% | 31.20% | 23.10% | 16.90% | 14.20% |
| | Moirai-small | 9.20% | 6.00% | 11.60% | 6.70% | 3.90% | 3.40% | 9.90% | 5.20% |
| | Moirai-base | 12.20% | 7.30% | 14.40% | 7.90% | 7.40% | 3.90% | 10.60% | 4.00% |
| | Moirai-large | 14.00% | 8.00% | 18.10% | 9.30% | 8.30% | 3.90% | 14.20% | 7.40% |
| | Chronos-tiny | 14.50% | 7.00% | 10.30% | 6.40% | 23.30% | 9.10% | 11.70% | 5.90% |
| **Data** | ETTh1 | 3.40% | 1.80% | 4.00% | 1.50% | 2.00% | 1.00% | 5.60% | 2.60% |
| | ETTh2 | 6.40% | 3.30% | 4.80% | 2.80% | 9.80% | 6.50% | 12.00% | 9.00% |
| | ETTm1 | 15.90% | 8.80% | 12.40% | 6.90% | 19.30% | 12.80% | 15.50% | 10.50% |
| | ETTm2 | 23.70% | 15.50% | 28.70% | 17.70% | 17.10% | 11.90% | 27.20% | 18.10% |
| | Electricity | 16.10% | 9.30% | 20.50% | 10.90% | 6.80% | 2.90% | 14.60% | 6.00% |
| | Exchange | 35.10% | 19.00% | 20.50% | 14.80% | 46.50% | 27.30% | 14.10% | 7.20% |
| | Traffic | 12.40% | 7.70% | 13.10% | 7.30% | 4.30% | 8.00% | 12.10% | 6.70% |
| | Weather | 1.50% | 1.10% | -1.30% | 0.40% | 4.10% | 1.50% | -0.40% | -0.20% |
| **Task** | 24 | 21.60% | 13.40% | 20.10% | 12.50% | 16.60% | 14.00% | 21.80% | 14.30% |
| | 48 | 14.00% | 8.30% | 13.60% | 7.90% | 12.80% | 8.90% | 12.30% | 7.30% |
| | 96 | 10.30% | 6.20% | 11.70% | 6.60% | 9.60% | 5.80% | 7.30% | 3.80% |
| | 192 | 12.30% | 6.10% | 9.10% | 5.20% | 19.40% | 9.10% | 9.40% | 4.70% |

From a data perspective, the improvement is most pronounced in the Exchange and ETTm2 datasets, exceeding 20%. However, the gains on the Weather dataset are relatively small, likely due to the inherent variability of weather data. From a task perspective, TATO demonstrates stable improvements across all prediction lengths, with shorter ones showing greater benefits. Additionally, the %Promotions for STD and IQR suggest a reduction in variance, highlighting TATO's ability to deliver more accurate and robust forecasts.

## A.7 MORE SCALABILITY EXPERIMENTS

We conduct a series of scaling experiments on different "number of samples", "number of trials", and "more diverse horizons". Compared to the experiments in our main text, we take more samples, more trials, and longer forecasting horizons into consideration. We also use **Time Cost** and **Loss Improvements** to evaluate the scaling process. For simplicity, all efficiency tests are conducted on Timer-LOTSA, and performance improvements are calculated on ETTh1 and ETTm1. Table 7 presents performance across different sample sizes and transformation trials. Table 8 presents the performance across different prediction horizons.

In terms of efficiency, the time cost shows a near-linear relationship with the number of samples, a trend that holds across different transformation trials. In terms of performance, TATO demonstrates robust results when the number of samples (or transformation trials) exceeds 500. The performance varies across horizons, and the required time is acceptable, further attesting to TATO's broad applicability.

Table 7: Scaling performance under different sample sizes and transformation trials. The results are averaged by horizons (24, 48, 96, 192, 336, 720).

| | **Trials** | | | **Samples** | | |
|---|---|---|---|---|---|---|
| **Metric** | 500 | 1000 | 1500 | 500 | 1000 | 1500 |
| Time Cost | 428.3 s | 491.1 s | 569.4 s | 428.3 s | 478.9 s | 503.4 s |
| MSE Imp. | 21.2 % | 22.3 % | 20.4 % | 21.2 % | 21.8 % | 20.6 % |
| MAE Imp. | 12.6 % | 13.2 % | 12.3 % | 12.6 % | 12.6 % | 12.1 % |

Table 8: Scaling performance across different prediction horizons. The improvement metric is defined as (MSE improvement percentage / MAE improvement percentage).

| Dataset | Mean Time(s) | Average Imp | Horizon 24 | Horizon 48 | Horizon 96 | Horizon 192 | Horizon 336 | Horizon 720 |
|---------|--------------|-------------|------------|------------|------------|-------------|-------------|-------------|
| ETTh1 | **411.1** | **9.5%/4.6%** | 19.4%/11.4% | -17.7%/-11.1% | 0.0%/0.0% | 0.0%/0.0% | 8.1%/4.3% | 28.9%/17.3% |
| ETTm1 | **445.5** | **23.3%/16.2%** | 67.3%/44.8% | 53.5%/32.6% | 39.6%/22.8% | 18.3%/9.3% | -0.5%/0.8% | 4.0%/2.6% |
| Exchange | **408.3** | **35.5%/31.3%** | 89.4%/67.7% | 81.5%/58.1% | 66.1%/44.2% | 35.0%/23.1% | -11.9%/1.3% | -26.1%/4.8% |
| Traffic | **305.1** | **2.0%/1.5%** | 3.6%/2.5% | 3.2%/1.6% | 1.8%/1.0% | 1.2%/1.3% | 1.4%/1.7% | 1.2%/1.3% |

