# OpenReview forum: "Adapt Data to Model: Adaptive Transformation Optimization for Domain-shared Time Series Foundation Models"
_ICLR.cc/2026/Conference — ICLR 2026 Poster_

### Official Review · Reviewer_8T1Q · 2025-10-26

**Soundness:** 3
**Presentation:** 4
**Contribution:** 2
**Rating:** 6
**Confidence:** 4

**Summary:**

This paper introduces Time-Series Adaptive Transformation Optimization (TATO), a framework designed to enhance the generalizability of frozen large time series models (LTMs) across diverse domains. Instead of retraining or creating new models, TATO automatically searches for empirically optimal transformation pipelines—comprising context slicing, scale normalization, and outlier correction—to adapt LTMs to new data distributions. Through a robust two-stage ranking mechanism, TATO ensures consistent performance across metrics and achieves substantial forecasting improvements, reducing MSE by up to 68.4% on standard benchmarks, all within a practical optimization time of under two minutes.

**Strengths:**

1.	This paper stands on a data manipulation perspective to improve the performance of time series forecasting, which is kind of novel.
2.	TATO is evaluated on both time series foundation models and LLM-based models to demonstrate its universal applicability across various model architectures.
3.	Performance gain brought by TATO is significant, reaching up to 68.4%.

**Weaknesses:**

1.	It is hard to understand what transformations are applied to the time series in Figure 1.
2.	TATO’s effectiveness heavily relies on a fixed set of nine transformation operators. The search space is manually defined, which may limit adaptability to unseen or highly domain-specific data types.
3.	The framework is presented primarily as an empirical optimization procedure, with little theoretical grounding on why certain transformations generalize well across domains or how the two-stage Pareto ranking ensures robustness.

**Questions:**

1.	Can TATO be applied to other time series tasks, such as imputation?
2.	What is the relationship between FrozenForecasting and Data2Model? It appears that the authors use them interchangeably.

---

> ### Author Response · Authors · 2025-11-23
> **Weakness 1 and 2**
>
> We sincerely thank Reviewer 8T1Q for the detailed review and insightful questions.
>
> > **W1:** "Explanation of Figure 1"
>
> The transformations shown in Figure 1 are just those presented in Appendix 3. Specifically, example 1 used Sampler at Eq. (2), example 2 used Eq. (7), and example 3 used Eq. (5). We will modify the figure in a later version of the paper.
>
>
> > **W2:** "Search space limitation"
>
> Yes, it is right that TATO relies on the search space of transformations. We manually define the search space based on practical experience, with efficiency in mind. **The primary contribution of TATO is an extensible framework for optimizing data to improve the forecasting performance of LTMs**. A more comprehensive search space will be explored in our future work.

---

> ### Author Response · Authors · 2025-11-23
> **Weakness 3 (Part I)**
>
> > **W3:** "Theoretical justification and deeper insights"
>
> In time series forecasting, the theoretical justification for the adaptation between model and data remains an unexplored area. Some recent efforts [1-2] have provided experimental analyses of the preferences of time-series forecasting models for data characteristics. However, **in most real-world datasets, the miscellaneous characteristics are not clearly defined, which limits the practicality of data characteristic analysis methods**. Our proposed method offers a novel approach to addressing the gap between data and models, providing more substantial evidence of models' preferences.
>
> [1] Wang F, Li Y, Shao Z, et al. ARIES: Relation Assessment and Model Recommendation for Deep Time Series Forecasting[J]. arXiv preprint arXiv:2509.06060, 2025.
>
> [2] Qiu X, Hu J, Zhou L, et al. TFB: Towards Comprehensive and Fair Benchmarking of Time Series Forecasting Methods[J]. Proceedings of the VLDB Endowment, 2024, 17(9): 2363-2377.
>
> Our experiments reveal that **optimal transformation strategies across datasets or LTMs differ significantly and have a significant impact on performance**.
>
> The best transformation strategies for different LTMs on ETTh1 for horizon 192 are compared as follows:
>
> | Model   | sampler | trimmer | aligner | normalizer | imputator | denoiser | order |
> |--------------|----------------|-----------------|--------------|-------------------|-----------------------|---------------|-----------------|
> | Timer-UTSD   | 1              | 1440            | -         | -              | 3_sigma               | -            | type 1        |
> | Timer-LOTSA  | 1              | 672             | -         | -              | -                  | -            | type 3        |
> | MOIRAI-small | 2              | 1344            | -         | robust            | -                  | -            | type 3        |
> | MOIRAI-base  | 2              | 864             | √   | -              | 3_sigma               | √            | type 3        |
> | MOIRAI-large | 1              | 1248            | -         | standard          | 1.5_iqr               | √            | type 1        |
> | Chronos-tiny | 2              | 480             | -         | robust            | -                  | √            | type 3        |
>
>
>
> The best transformation strategies for different LTMs on ETTm1 for horizon 192 are compared as follows:
>
>
> | Model   | sampler | trimmer | aligner | normalizer | imputator | warper | differentiator | denoiser | order |
> |--------------|----------------|-----------------|--------------|-------------------|-----------------------|---------------|------------------|-----------------|---------------|
> | Timer-UTSD   | 1              | 960             | √   | standard          | -                  | log           | -                | √            | type 1        |
> | Timer-LOTSA  | 1              | 1440            | -         | standard          | -                  | -          | √                | √            | type 3        |
> | MOIRAI-small | 2              | 1248            | -         | standard          | 3_sigma               | -          | -                | -            | type 1        |
> | MOIRAI-base  | 2              | 1440            | √   | -              | 3_sigma               | log           | -                | -            | type 1        |
> | MOIRAI-large | 2              | 1440            | -         | robust            | 1.5_iqr               | -          | -                | -            | type 1        |
> | Chronos-tiny | 2              | 768             | √   | robust            | -                  | log           | -                | -            | type 3        |
>
> The best transformation strategies for different LTMs on Traffic for horizon 192 are compared as follows:
>
> | Model   | sampler | trimmer | aligner | normalizer | imputator | warper | order |
> |--------------|----------------|-----------------|--------------|-------------------|-----------------------|---------------|---------------|
> | Timer-UTSD   | 1              | 576             | √   | standard          | 1.5_iqr               | -          | type 3        |
> | Timer-LOTSA  | 1              | 1056            | -         | -              | 3_sigma               | -          | type 1        |
> | MOIRAI-small | 1              | 1344            | √   | -              | -                  | log           | type 3        |
> | MOIRAI-base  | 1              | 864             | -         | standard          | 1.5_iqr               | -          | type 1        |
> | MOIRAI-large | 1              | 1440            | -         | -              | 3_sigma               | -          | type 1        |
> | Chronos-tiny | 1              | 576             | √   | -              | -    | -          | type 3        |

---

> ### Author Response · Authors · 2025-11-23
> **Weakness 3 (Part II)**
>
> By comparing models on the same datasets, it is noteworthy that different LTMs exhibit distinct preferences for transformations, further strengthening the analysis of the gap between data and models in a novel perspective.
>
> We also summarize the above results in the view of LTMs. The best transformation strategies for the above three datasets by Timer-UTSD for horizon 192 are compared as follows:
>
> | Dataset  | sampler | trimmer | aligner | normalizer | imputator | warper |denoiser | order |
> |-----------|----------------|-----------------|--------------|-------------------|-----------------------|---------------|-----------------|---------------|
> | ETTh1     | 1              | 1440            | -         | -              | 3_sigma               | -          | -            | type 1        |
> | ETTm1     | 1              | 960             | √   | standard          | -                  | log           | √            | type 1        |
> | Traffic   | 1              | 576             | √   | standard          | 1.5_iqr               | -          | -            | type 3        |
>
> The best transformation strategies for the above three datasets by MOIRAI-large for horizon 192 are compared as follows:
>
> | Dataset | sampler | trimmer  | normalizer | imputator |  denoiser | order |
> |-----------|-----------------|--------------|-------------------|-----------------------|-----------------|---------------|
> | ETTh1     | 1              | 1248          | standard          | 1.5_iqr               | √            | type 1        |
> | ETTm1     | 2              | 1440        | robust            | 1.5_iqr               | -            | type 1        |
> | Traffic   | 1              | 1440    | -              | 3_sigma               | -            | type 1        |
>
> From the above results regarding LTMs, we can see that **Timer-UTSD prefers shorter inputs when dealing with more complex datasets, such as ETTm1 and Traffic. And for MOIRAI-large, long inputs are always preferred across datasets**.

---

> ### Author Response · Authors · 2025-11-23
> **Question 1 and 2**
>
> > **Q1:** "TATO for imputation"
>
>
> Thank you for the valuable question. We think TATO can be applied to other tasks. It mainly depends on the given LTMs.
>
> Notably, the transformations should be redesigned for imputation. For imputation, the output is a complement of the input. So **the context-related transformations, such as trimming, should not be used**. Besides, part of the input in imputation is masked rather than a real value. So **the value-related transformations, such as normalization and differentiation, need to take the mask into account**.
>
>
> > **Q2:** "Relationship between FrozenForecasting and Data2Model"
>
>
> FrozenForecasting, as we present in Definition 1 in Section 3.1, is a practical setting to use LTM facing multiple domains. In comparison, **Data2Model is a paradigm for improving LTMs' performance in FrozenForecasting**.
>
>
> We fully acknowledge your insightful comments.

---

### Official Review · Reviewer_kQ8y · 2025-10-30

**Soundness:** 3
**Presentation:** 2
**Contribution:** 3
**Rating:** 6
**Confidence:** 4

**Summary:**

This paper introduces an interesting paradigm that aligns time series data with a pretrained large time series model through transformation pipelines. By automatically selecting feasible pipelines via a two-stage ranking process, the proposed framework aims to enhance forecasting performance on target domains without requiring fine-tuning of the backbone models.

**Strengths:**

- The idea of leveraging transformation pipelines to align time series data with forecasting models is novel and promising, as it improves domain-adaptive forecasting performance while keeping the backbone models fixed.
- The proposed pipeline optimization process is efficient, typically requiring less than two minutes in most cases.
- Experimental results demonstrate that the proposed framework effectively boosts the performance of various backbone models on widely used benchmark datasets.

**Weaknesses:**

- Clarity and presentation:
  - The framework description lacks sufficient detail. For instance, how are individual trials sampled from the pipeline space? Providing pseudocode or an algorithmic overview would greatly improve readability and reproducibility.
  - It is mentioned that only a portion of the training samples are used in the TATO framework, but the selection strategy for these samples is unclear. A detailed explanation of how these samples are chosen from the full training set would strengthen the paper.
- Experimental analysis:
  - Table 2 reports runtime comparisons across different configurations, but it would be helpful to also include forecasting performance metrics in the same table to illustrate the efficiency–effectiveness tradeoff of TATO.
  - Although TATO is primarily proposed for large time series models, it would be interesting to investigate whether conventional models such as PatchTST can also benefit from TATO under cross-domain forecasting scenarios.
  - Section 4.5 compares LTMs tuned over eight datasets against TATO. However, it would be insightful to include results where the model is tuned on a *single* dataset (e.g., tuning Timer on ETTh1 with a comparable computational budget in terms of training time or samples) to better contextualize TATO’s performance.

**Questions:**

Please refer to the weaknesses part.

---

> ### Author Response · Authors · 2025-11-23
> **Weakness 1 and 2**
>
> Many thanks to Reviewer kQ8y for the detailed review and insightful questions.
>
> > **W1:** "Pseudocode"
>
> Thank you for the suggestion. We complement a detailed pseudocode in Appendix 2.3 as shown in the revised paper.
>
> > **W2:** "Sampling methods"
>
>
> Thank you for the insightful question.
>
> We conduct additional experiments to show the impact of different sampling methods. We implement two contrasting sampling protocols. The "random-samples" protocol conducts adaptation on randomly selected segments from the training dataset, whereas the "latest-samples" protocol exclusively uses the segment of the training time series that immediately precedes the test set. The input is fixed at 672. And the results are averaged across models (Timer-UTSD, Timer-LOTSA) and horizons (24, 48, 96, 192).
>
> | Dataset| **Random Samples mse/mae** | **Latest Samples mse/mae** |
> |-----------------------|----------------------------|--------------------------|
> | ETTh1                 | **0.3848/0.4733**          | 0.4032/0.4800            |
> | ETTh2                 | **0.3182/0.4342**          | 0.3451/0.4568            |
> | ETTm1                 | 0.2801/0.3867              | **0.2489/0.3627**        |
> | ETTm2                 | 0.4421/0.4885              | **0.3624/0.4297**        |
> | Electricity           | **0.1743/0.2879**          | 0.1848/0.3015            |
> | Exchange              | 0.3232/0.3979              | **0.2564/0.3648**        |
> | Traffic               | **0.0593/0.1359**          | 0.0610/0.1393            |
> | Weather               | 0.6046/**0.5278**          | **0.5837**/0.5374        |
> | Mean                  | 0.3233/0.3915              | **0.3057/0.3840**        |
>
> In the paper, we applied the "random-samples" protocol. But on average, **"latest-samples" performs better than "random-samples", primarily because it is closer to the test set distribution**. However, "latest-samples" has lower diversity in sampling, increasing the risk of overfitting during adaptation and potentially hindering its ability to capture diverse data characteristics.

---

> ### Author Response · Authors · 2025-11-23
> **Weakness 3**
>
> > **W3:** "Performance in efficiency experiments"
>
>
> We present experiments on the ETTh1 and Electricity datasets with varying numbers of trials and samples to illustrate the underlying trade-off between efficiency and effectiveness. (Horizon=96)
>
> Different Trials with the Samples=100:
>
> | Trials | Timer-UTSD mse_imp / time | Moirai-base mse_imp / time | Chronos-tiny mse_imp / time |
> |---------------------|--------------------------------|----------------------------|-----------------------------|
> | 25                  | 0.0% / 11.3s                   | 8.3% / 3.6s                | -2.8% / 21.6s               |
> | 100                 | 0.7% / 46.7s                   | 14.6% / 14.7s              | -0.6% / 99.5s               |
> | 500                 | 3.0% / 237.1s                  | 14.8% / 93.5s              | -2.1% / 734.5s              |
>
>
> Different Samples with the Trials=100:
>
> | Samples | Timer-UTSD mse_imp / time | Moirai-base mse_imp / time | Chronos-tiny mse_imp / time |
> |----------------------|--------------------------------|----------------------------|-----------------------------|
> | 25                   | 0.8% / 46.8s                   | 5.0% / 9.5s                | -3.9% / 70.8s               |
> | 100                  | 0.7% / 46.7s                   | 14.6% / 14.7s              | -0.6% / 99.5s               |
> | 500                  | 2.7% / 54.1s                   | 13.5% / 54.9s              | -1.7% / 418.5s              |
>
>
> The results show that **increasing the number of samples and trials leads to greater time overhead, while model performance shows a commensurate improvement trend**. Consequently, a hyperparameter search is preferred in practice to achieve a desirable balance between efficiency and effectiveness.

---

> ### Author Response · Authors · 2025-11-23
> **Weakness 4**
>
> > **W4:** "Benefits for PatchTST"
>
>
> We first use TATO to select a set of data transformations and apply them to the PatchTST training data. Hyperparameter optimization is conducted on both approaches. The results show that **TATO commonly provided better transformation strategies for PatchTST** than the original default way.
>
> | Dataset | Setting | vanilla PatchTST（MSE/MAE） | PatchTST+TATO (MSE/MAE) |
> | ------- | ------- | --------------------------------- | ---------------------------- |
> | ETTh1   | 672-24  | 0.0443/0.1588                     | **0.0317/0.1356**      |
> | ETTh1   | 672-48  | 0.0615/0.1952                     | **0.0515/0.1785**      |
> | ETTh1   | 672-96  | 0.0734/0.2113                     | **0.0613/0.1937**      |
> | ETTh1   | 672-192 | 0.1324/0.2847                     | **0.0859/0.2323**      |
> | Traffic | 672-24  | 0.1211/0.2098                     | **0.1174/0.2047**      |
> | Traffic | 672-48  | 0.1273/0.2174                     | **0.1269/0.2169**      |
> | Traffic | 672-96  | 0.1325/0.2207                     | **0.1297/0.2159**      |
> | Traffic | 672-192 | **0.1381/0.2207**                 | 0.1381/0.2269      |

---

> ### Author Response · Authors · 2025-11-23
> **Weakness 5**
>
> > **W5:** "Comparison with finetuning on each single dataset"
>
> We first conduct an additional comparison between TATO and finetuning on each dataset to provide a more comprehensive understanding of TATO's efficacy.
>
> In finetuning experiments, we use the default implementation from "Timer: Transformers for Time Series Analysis at Scale" and keep the finetune settings and hyperparameters aligned with their official code release. The samples to be used for finetuning are the same as those needed by TATO for fairness (500 samples). The finetuned checkpoints are applied to the same testing logic aligned to TATO.
>
> - Effectiveness:
>     - **In most cases, TATO reaches similar or better performance compared to finetuning (on each dataset)**.
>     - For smaller datasets such as ETT and Exchange, TATO consistently outperforms finetuning.
>     - For larger datasets such as Traffic, Weather, and Electricity, under a higher diversity of the samples, the advantage of TATO reduces, but it still achieves similar results.
> - Efficiency:
>   - Time cost: The timespan of the finetuning process is highly unstable due to different dataloading parallel techniques and dataset size. In our experiment, fine-tuning Timer typically takes about 300 seconds. TATO with 500 trials takes 76 seconds on Timer, which is about **25\% of the finetuning timespan**.
>   - GPU memory: The finetuning of Timer requires at least 4GB of GPU memory, while TATO on Timer only requires 1GB, just **25\% of the finetuning GPU memory**.
>
> | MSE/MAE     | Timer-UTSD-finetune | Timer-UTSD-TATO | Timer-LOTSA-finetune | Timer-LOTSA-TATO |
> | ----------- | ------------- | --------------- | -------------- | ---------------- |
> | Electricity | 0.177/0.290   | **0.174/0.287**     | 0.1705/**0.287**   | **0.1703**/0.289     |
> | ETTh1       | 0.419/0.492   | **0.384/0.473**     | 0.443/0.506    | **0.411/0.484**      |
> | ETTh2       | 0.338/0.449   | **0.318/0.434**     | 0.382/0.487    | **0.352/0.463**      |
> | ETTm1       | 0.302/0.401   | **0.280/0.386**     | 0.377/0.461    | **0.248/0.356**      |
> | ETTm2       | 0.478/0.529   | **0.442/0.488**     | 0.317/0.420    | **0.249/0.352**      |
> | Exchange    | 0.522/0.499   | **0.323/0.397**     | 0.846/0.712    | **0.231/0.344**      |
> | Traffic     | **0.058/0.132**   | 0.059/0.135     | 0.0615/**0.138**   | **0.0614**/0.140     |
> | Weather     | **0.519/0.508**   | 0.604/0.527     | 0.608/0.562    | **0.593/0.542**      |
> | **AVERAGE**     | 0.352/0.413   | **0.323/0.391**     | 0.400/0.447    | **0.289/0.371**      |
>
> Furthermore, we conduct additional experiments to show how TATO can improve finetuning. We use TATO to select a set of data transformations, then apply them for finetuning.
>
> | Dataset | Setting  (input and output) | vanilla Timer-UTSD（MSE/MAE） | Timer-UTSD+TATO (MSE/MAE) |
> | ------- | ------- | --------------------------------- | ---------------------------- |
> | ETTh1   | 672-24  | **0.0319/0.1352**                     | 0.0333/0.1414                |
> | ETTh1   | 672-48  | 0.0530/0.1787                     | **0.0465/0.1693**                |
> | ETTh1   | 672-96  | 0.0776/0.2228                     | **0.0547/0.1821**                |
> | ETTh1   | 672-192 | 0.0815/0.2302                     | **0.0747/0.2151**               |
> | Traffic | 672-24  | 0.0981/0.1805                     | **0.0884/0.1649**                |
> | Traffic | 672-48  | 0.1017/0.1849                     | **0.0985/0.1719**                |
> | Traffic | 672-96  | 0.1053/0.1809                     | **0.1033/0.1741**                |
> | Traffic | 672-192 | 0.1341/0.2152                     | **0.1336/0.2079**                |
>
> The experiments demonstrate that TATO can provide a better strategy for data transformations, offering a new way to use TATO.
>
>
> We fully acknowledge your insightful comments.

---

> > ### Comment · Reviewer_kQ8y · 2025-11-25
> >
> > Thank you for your detailed rebuttal, which helps to address my concerns. I will raise my ratings.

---

### Official Review · Reviewer_YMH3 · 2025-10-31

**Soundness:** 2
**Presentation:** 3
**Contribution:** 3
**Rating:** 4
**Confidence:** 4

**Summary:**

The paper proposes TATO, a framework to adapt data rather than models for time series forecasting. Instead of fine tuning large time series models (LTMs), TATO keeps them frozen and learns optimal preprocessing transformations to improve performance across domains. The method searches transformation pipelines including context slicing, scale normalization, and outlier correction using hyperparameter optimization and a two stage Pareto based ranking for robustness. Experiments on LTMs (e.g., Timer, Moirai, Chronos) and datasets (ETT, Electricity, Exchange, Traffic, Weather) show consistent improvements. This work introduces a practical FrozenForecasting paradigm for universal, efficient forecasting.

**Strengths:**

The paper presents an original paradigm that shifts the focus from model adaptation to data adaptation, introducing the concept of FrozenForecasting for large time series models. The proposed TATO framework is well designed, combining hyperparameter optimization and Pareto based ranking to ensure both robustness and efficiency. The methodology is clearly described with experiments across several state of the art LTMs and diverse datasets. The work provides some evidence of performance improvement with minimal computational overhead, highlighting its practicality for real world applications.

**Weaknesses:**

While the idea of adapting data instead of models is novel, the paper could benefit from a deeper theoretical justification of why certain transformations consistently enhance generalization across domains. The search space for transformation pipelines, though compact, may still be computationally demanding for larger datasets or longer horizons, and scalability analyses beyond 500 samples are limited. Some test-time adaptation methods for time-series forecasting were completely ignored by the authors, such as TAFAS (Kim et al. (2025).) [1], PETSA (Medeiros et al. (2025)) [2], DynaTTA (Grover & Etemad. (2025)) [3], I would suggest the authors to review these additional methods for the related works to strong contribution of the proposed TATO, and also compare with some of the methods if possible, because in the main comparisson the paper is missing comparisson with other methods. Additionally, the ablation results could include more quantitative discussion on how each operator contributes to generalization rather than overall MSE reduction.



[1] HyunGi Kim, Siwon Kim, Jisoo Mok, and Sungroh Yoon. Battling the non-stationarity in time
series forecasting via test-time adaptation. In AAAI, pp. 17868–17876, 2025. URL https:
//doi.org/10.1609/aaai.v39i17.33965.

[2] Heitor Medeiros, Hossein Sharifi-Noghabi, Gabriel Oliveira, and Saghar Irandoust. Accurate
parameter-efficient test-time adaptation for time series forecasting. In Second Workshop on Test-
Time Adaptation: Putting Updates to the Test!, 06 2025. doi: 10.48550/arXiv.2506.23424.

[3] Shivam Grover and Ali Etemad. Shift-aware test time adaptation and benchmarking for time-series
forecasting. In Second Workshop on Test-Time Adaptation: Putting Updates to the Test! at ICML
2025, 2025.

**Questions:**

1. How sensitive is the optimization process to the size and diversity of the sampled data? Would using fewer or noisier samples significantly affect the quality of the selected transformation pipeline?

2. Could the authors clarify how TATO generalizes to multivariate or irregularly sampled time series, where contextual or scale transformations may interact differently?

3. How does TATO compare empirically or conceptually with recent test time adaptation methods TAFAS, PETSA, DynaTTA?

4. Are there any insights into which specific transformations contribute most to improving generalization across domains?

5. Could TATO be combined with lightweight fine tuning or LoRA style adaptation to further improve performance while maintaining efficiency, and if so, what trade offs might emerge?

---

> ### Author Response · Authors · 2025-11-23
> **Weakness and Question 4 (Part I)**
>
> We sincerely thank Reviewer YMH3 for providing valuable feedback and suggestions.
>
> > **W(1) & Q4:** "Theoretical justification and deeper insights, together with transformation contribution analysis"
>
> In time series forecasting, the theoretical justification for the adaptation between model and data remains an unexplored area. Some recent efforts [1-2] have provided experimental analyses of the preferences of time-series forecasting models for data characteristics. However, **in most real-world datasets, the miscellaneous characteristics are not clearly defined, which limits the practicality of data characteristic analysis methods**. Our proposed method offers a novel approach to addressing the gap between data and models, providing more substantial evidence of models' preferences.
>
> [1] Wang F, Li Y, Shao Z, et al. ARIES: Relation Assessment and Model Recommendation for Deep Time Series Forecasting[J]. arXiv preprint arXiv:2509.06060, 2025.
>
> [2] Qiu X, Hu J, Zhou L, et al. TFB: Towards Comprehensive and Fair Benchmarking of Time Series Forecasting Methods[J]. Proceedings of the VLDB Endowment, 2024, 17(9): 2363-2377.
>
> Our experiments reveal that **optimal transformation strategies across datasets or LTMs differ significantly and have a significant impact on performance**.
>
> The best transformation strategies for different LTMs on ETTh1 for horizon 192 are compared as follows:
>
> | Model   | sampler | trimmer | aligner | normalizer | imputator | denoiser | order |
> |--------------|----------------|-----------------|--------------|-------------------|-----------------------|---------------|-----------------|
> | Timer-UTSD   | 1              | 1440            | -         | -              | 3_sigma               | -            | type 1        |
> | Timer-LOTSA  | 1              | 672             | -         | -              | -                  | -            | type 3        |
> | MOIRAI-small | 2              | 1344            | -         | robust            | -                  | -            | type 3        |
> | MOIRAI-base  | 2              | 864             | √   | -              | 3_sigma               | √            | type 3        |
> | MOIRAI-large | 1              | 1248            | -         | standard          | 1.5_iqr               | √            | type 1        |
> | Chronos-tiny | 2              | 480             | -         | robust            | -                  | √            | type 3        |
>
>
>
> The best transformation strategies for different LTMs on ETTm1 for horizon 192 are compared as follows:
>
>
> | Model   | sampler | trimmer | aligner | normalizer | imputator | warper | differentiator | denoiser | order |
> |--------------|----------------|-----------------|--------------|-------------------|-----------------------|---------------|------------------|-----------------|---------------|
> | Timer-UTSD   | 1              | 960             | √   | standard          | -                  | log           | -                | √            | type 1        |
> | Timer-LOTSA  | 1              | 1440            | -         | standard          | -                  | -          | √                | √            | type 3        |
> | MOIRAI-small | 2              | 1248            | -         | standard          | 3_sigma               | -          | -                | -            | type 1        |
> | MOIRAI-base  | 2              | 1440            | √   | -              | 3_sigma               | log           | -                | -            | type 1        |
> | MOIRAI-large | 2              | 1440            | -         | robust            | 1.5_iqr               | -          | -                | -            | type 1        |
> | Chronos-tiny | 2              | 768             | √   | robust            | -                  | log           | -                | -            | type 3        |
>
> The best transformation strategies for different LTMs on Traffic for horizon 192 are compared as follows:
>
> | Model   | sampler | trimmer | aligner | normalizer | imputator | warper | order |
> |--------------|----------------|-----------------|--------------|-------------------|-----------------------|---------------|---------------|
> | Timer-UTSD   | 1              | 576             | √   | standard          | 1.5_iqr               | -          | type 3        |
> | Timer-LOTSA  | 1              | 1056            | -         | -              | 3_sigma               | -          | type 1        |
> | MOIRAI-small | 1              | 1344            | √   | -              | -                  | log           | type 3        |
> | MOIRAI-base  | 1              | 864             | -         | standard          | 1.5_iqr               | -          | type 1        |
> | MOIRAI-large | 1              | 1440            | -         | -              | 3_sigma               | -          | type 1        |
> | Chronos-tiny | 1              | 576             | √   | -              | -    | -          | type 3        |

---

> ### Author Response · Authors · 2025-11-23
> **Weakness and Question 4 (Part II)**
>
> By comparing models on the same datasets, it is noteworthy that **different LTMs exhibit distinct preferences for transformations**, further strengthening the analysis of the gap between data and models in a novel perspective.
>
> We also summarize the above results in the view of LTMs. The best transformation strategies for the above three datasets by Timer-UTSD for horizon 192 are compared as follows:
>
> | Dataset  | sampler | trimmer | aligner | normalizer | imputator | warper |denoiser | order |
> |-----------|----------------|-----------------|--------------|-------------------|-----------------------|---------------|-----------------|---------------|
> | ETTh1     | 1              | 1440            | -         | -              | 3_sigma               | -          | -            | type 1        |
> | ETTm1     | 1              | 960             | √   | standard          | -                  | log           | √            | type 1        |
> | Traffic   | 1              | 576             | √   | standard          | 1.5_iqr               | -          | -            | type 3        |
>
> The best transformation strategies for the above three datasets by MOIRAI-large for horizon 192 are compared as follows:
>
> | Dataset | sampler | trimmer  | normalizer | imputator |  denoiser | order |
> |-----------|-----------------|--------------|-------------------|-----------------------|-----------------|---------------|
> | ETTh1     | 1              | 1248          | standard          | 1.5_iqr               | √            | type 1        |
> | ETTm1     | 2              | 1440        | robust            | 1.5_iqr               | -            | type 1        |
> | Traffic   | 1              | 1440    | -              | 3_sigma               | -            | type 1        |
>
> From the above results regarding LTMs, we can see that **Timer-UTSD prefers shorter inputs when dealing with more complex datasets**, such as ETTm1 and Traffic. And **for MOIRAI-large, longer inputs are always preferred across datasets**.

---

> ### Author Response · Authors · 2025-11-23
> **Weakness on Scalability**
>
> > **W(2):** "Scalability"
>
> We conduct a series of experiments on different "number of samples",  "number of trials", and "more diverse horizons". All these experiments are conducted upon Timer-LOTSA.
>
> 1. Results on different numbers of samples, averaged by horizons (24,48,96,192,336,720) and datasets (ETTh1, ETTm1):
>
> | Num of samples | **time cost (s)** | **mse_imp** | **mae_imp** |
> |------------------|-------------------|-------------|-------------|
> | **500**      | 428.3             | 21.2%       | 12.6%       |
> | **1000**    | 478.9             | 21.8%       | 12.6%       |
> | **1500**    | 503.4             | 20.6%       | 12.1%       |
>
> 2. Results on different transformation trials, averaged by horizons (24,48,96,192,336,720):
>
> | Num of Trials  | **time cost (s)** | **mse_imp** | **mae_imp** |
> |-----------------|-------------------|-------------|-------------|
> | **500**     | 428.3             | 21.2%       | 12.6%       |
> | **750**       | 491.1             | 22.3%       | 13.2%       |
> | **1000**   | 569.4             | 20.4%       | 12.3%       |
>
> 3. Results on different horizons (Timer-LOTSA), with num of samples 500 and num of Trials 500:
>
> | **Dataset** | **mean time cost (s)** | **horizons' avg mse/mae_imp(%) (24,48,96,192,336,720)** | **mse/mae_imp(%)-24** | **mse/mae_imp(%)-48** | **mse/mae_imp(%)-96** | **mse/mae_imp(%)-192** | **mse/mae_imp(%)-336** | **mse/mae_imp(%)-720** |
> |-----------|------------------------|-------------------------|-----------------------|-----------------------|-----------------------|------------------------|------------------------|------------------------|
> | ETTh1     | **411.1**         | **12.2/6.7**      | 20.3/11.3             | 8.7/3.3               | 0.0/0.0               | 0.0/0.0                | 3.6/2.1                | 40.6/23.0              |
> | ETTm1     | **445.5**          | **30.2/18.6**          | 67.4/44.9             | 53.4/32.6             | 39.5/22.7             | 18.2/9.3               | 1.0/1.8                | 1.3/0.4                |
> | Exchange  | **408.3**           | **41.7/34.2**     | 88.5/66.3             | 80.9/57.3             | 69.2/46.4             | 52.2/31.1              | 21.6/10.6              | -62.2/-6.8             |
> | Traffic   | **305.1**           | **1.1/1.1**         | 0.0/0.0               | 0.0/0.0               | 1.9/1.2               | 1.3/1.6                | 1.0/1.3                | 2.2/2.5                |
>
>
> In terms of efficiency, the time cost shows a near-linear relationship with the number of samples, a trend that holds across different transformation trials.
>
> In terms of performance, TATO demonstrates robust results when the number of samples (or transformation trials) exceeds 500. The performance varies across horizons, and the required time is acceptable, further attesting to TATO's broad applicability.

---

> ### Author Response · Authors · 2025-11-23
> **Weakness and Question 3**
>
> > **W(3) & Q3:** "Comparison with test-time adaptation"
>
>
>
> Test-time adaptation methods also allow data to be adapted under a predefined assumption about data characteristics. We conduct additional experiments comparing with test-time adaptation. **Unlike TATO, existing test-time adaptation methods mainly address forecasting issues for conventional deep learning models rather than LTMs**. Under consistent experimental settings with TATO, we evaluated two test-time adaptation approaches (DynaTTA and TAFAS) for Timer-LOTSA and Timer-UTSD models on the datasets ETTh1 and Electricity.
>
> In general, **TATO achieves more substantial performance improvements and demonstrates greater data-adaptation capability**. However, both test-time adaptation approaches perform poorly across the two LTMs. We will conduct more experiments with comprehensive comparisons to test-time adaptation in the final version of the paper.
>
> |     Model     | Dataset | TATO_mse_imp | DynaTTA_mse_imp | TAFAS_mse_imp |
> | :-------------: | :------: | :----------: | :-------------: | :-----------: |
> | **Timer-LOTSA** |  ETTh1   |   **7.3%**   |      -8.2%      |     -0.3%     |
> |                 |   ECL    |     0.0%     |      -1.8%      |   **0.2%**    |
> | **Timer-UTSD**  |  ETTh1   |   **5.1%**   |     -18.2%      |     0.0%      |
> |                 |   ECL    |   **0.8%**   |      -3.7%      |     0.0%      |

---

> ### Author Response · Authors · 2025-11-23
> **Question 1**
>
> > **Q1:** "Sensitivity to the size and diversity of the sampled data"
>
> In the paper, we use Figures 4 and 5 to present experiments on TATO's sensitivity to different transformations, components, and the number of trials and samples.
>
> In Figure 4, we see that the proposed steps, **"Augmentation" and "TwoStageRank", play significant roles in TATO's robustness. They remarkably reduce performance variation**, making TATO less sensitive to the size and diversity of the sampled data.
>
> We conduct additional experiments to show the impact of different sampling methods. We implement two contrasting sampling protocols. The "random-samples" protocol conducts adaptation on randomly selected segments from the training dataset, whereas the "latest-samples" protocol exclusively uses the segment of the training time series that immediately precedes the test set. The input is fixed at 672. And the results are averaged across models (Timer-UTSD, Timer-LOTSA) and horizons (24, 48, 96, 192).
>
> | Dataset| **Random Samples mse/mae** | **Latest Samples mse/mae** |
> |-----------------------|----------------------------|--------------------------|
> | ETTh1                 | **0.3848/0.4733**          | 0.4032/0.4800            |
> | ETTh2                 | **0.3182/0.4342**          | 0.3451/0.4568            |
> | ETTm1                 | 0.2801/0.3867              | **0.2489/0.3627**        |
> | ETTm2                 | 0.4421/0.4885              | **0.3624/0.4297**        |
> | Electricity           | **0.1743/0.2879**          | 0.1848/0.3015            |
> | Exchange              | 0.3232/0.3979              | **0.2564/0.3648**        |
> | Traffic               | **0.0593/0.1359**          | 0.0610/0.1393            |
> | Weather               | 0.6046/**0.5278**          | **0.5837**/0.5374        |
> | Mean                  | 0.3233/0.3915              | **0.3057/0.3840**        |
>
> In the paper, we applied the "random-samples" protocol. But on average, **"latest-samples" performs better than "random-samples", primarily because it is closer to the test set distribution**. However, "latest-samples" has lower diversity in sampling, increasing the risk of overfitting during adaptation and potentially hindering its ability to capture diverse data characteristics.

---

> ### Author Response · Authors · 2025-11-23
> **Question 2**
>
> > **Q2:** "Multivariate forecasting"
>
> Thank you for this very insightful question. Let us suppose that the concerned multivariate forecasting includes one target variate and several covariates. In multivariate forecasting, we apply TATO via a context-and-scope-coupling approach. First, a complete transformation optimization is applied to the target variable. Then, the context- and scope-related transformations are repeatedly used to adapt the covariates, thereby vastly reducing the covariate search space.
>
> It is reasonable to reuse the context- and scope-related transformations of the target variate.
> - Context-related transformations, such as trimming, will change the time points selected as input. **In multivariate forecasting, dependencies among variables should be modeled intuitively over the same candidate time points**.
> - Scope-related transformations, such as normalization, will change the value scope of variables. **To maintain the relative value dependencies between variables, the scope-related transformations should be the same**.

---

> ### Author Response · Authors · 2025-11-23
> **Question 5**
>
> > **Q5:** "Combination with finetuning"
>
> Thank you for the very insightful question. We first conduct an additional comparison between TATO and finetuning on each dataset to provide a more comprehensive understanding of TATO's efficacy.
>
> In finetuning experiments, we use the default implementation from "Timer: Transformers for Time Series Analysis at Scale" and keep the finetune settings and hyperparameters aligned with their official code release. The samples to be used for finetuning are the same as those needed by TATO for fairness (500 samples). The finetuned checkpoints are applied to the same testing logic aligned to TATO.
>
> - Effectiveness:
>     - **In most cases, TATO reaches similar or better performance compared to finetuning (on each dataset)**.
>     - For smaller datasets such as ETT and Exchange, TATO consistently outperforms finetuning.
>     - For larger datasets such as Traffic, Weather, and Electricity, under a higher diversity of the samples, the advantage of TATO reduces, but it still achieves similar results.
> - Efficiency:
>   - Time cost: The timespan of the finetuning process is highly unstable due to different dataloading parallel techniques and dataset size. In our experiment, fine-tuning Timer typically takes about 300 seconds. TATO with 500 trials takes 76 seconds on Timer, which is about **25\% of the finetuning timespan**.
>   - GPU memory: The finetuning of Timer requires at least 4GB of GPU memory, while TATO on Timer only requires 1GB, just **25\% of the finetuning GPU memory**.
>
> | MSE/MAE     | Timer-UTSD-finetune | Timer-UTSD-TATO | Timer-LOTSA-finetune | Timer-LOTSA-TATO |
> | ----------- | ------------- | --------------- | -------------- | ---------------- |
> | Electricity | 0.177/0.290   | **0.174/0.287**     | 0.1705/**0.287**   | **0.1703**/0.289     |
> | ETTh1       | 0.419/0.492   | **0.384/0.473**     | 0.443/0.506    | **0.411/0.484**      |
> | ETTh2       | 0.338/0.449   | **0.318/0.434**     | 0.382/0.487    | **0.352/0.463**      |
> | ETTm1       | 0.302/0.401   | **0.280/0.386**     | 0.377/0.461    | **0.248/0.356**      |
> | ETTm2       | 0.478/0.529   | **0.442/0.488**     | 0.317/0.420    | **0.249/0.352**      |
> | Exchange    | 0.522/0.499   | **0.323/0.397**     | 0.846/0.712    | **0.231/0.344**      |
> | Traffic     | **0.058/0.132**   | 0.059/0.135     | 0.0615/**0.138**   | **0.0614**/0.140     |
> | Weather     | **0.519/0.508**   | 0.604/0.527     | 0.608/0.562    | **0.593/0.542**      |
> | **AVERAGE**     | 0.352/0.413   | **0.323/0.391**     | 0.400/0.447    | **0.289/0.371**      |
>
> Furthermore, we conduct additional experiments to show how TATO can improve finetuning. We use TATO to select a set of data transformations, then apply them for finetuning.
>
> | Dataset | Setting (input and output) | vanilla Timer-UTSD (MSE/MAE） | Timer-UTSD+TATO (MSE/MAE) |
> | ------- | ------- | --------------------------------- | ---------------------------- |
> | ETTh1   | 672-24  | **0.0319/0.1352**                     | 0.0333/0.1414                |
> | ETTh1   | 672-48  | 0.0530/0.1787                     | **0.0465/0.1693**                |
> | ETTh1   | 672-96  | 0.0776/0.2228                     | **0.0547/0.1821**                |
> | ETTh1   | 672-192 | 0.0815/0.2302                     | **0.0747/0.2151**               |
> | Traffic | 672-24  | 0.0981/0.1805                     | **0.0884/0.1649**                |
> | Traffic | 672-48  | 0.1017/0.1849                     | **0.0985/0.1719**                |
> | Traffic | 672-96  | 0.1053/0.1809                     | **0.1033/0.1741**                |
> | Traffic | 672-192 | 0.1341/0.2152                     | **0.1336/0.2079**                |
>
> The experiments demonstrate that TATO can provide a better strategy for data transformations, offering a new way to use TATO.
>
> We fully acknowledge your insightful comments.

---

> > ### Comment · Reviewer_YMH3 · 2025-11-28
> >
> > Dear authors, thank you for the detailed rebuttal; it clarified my questions. I hope to see the points discussed here in the revised version of the main manuscript if the paper gets accepted. I am raising my scores after considering the effort on the rebuttal.

---

> > > ### Author Response · Authors · 2025-11-28
> > >
> > > Thank you, Reviewer YMH3, for the insightful review and acknowledgement. We have included most of the points from the rebuttal in the appendix. We will revise them in a more compact and unified way in the final version.

---

> > > > ### Comment · Reviewer_YMH3 · 2025-11-28
> > > >
> > > > Thanks for the clarification.
> > > >
> > > > Do you have any justification for why you selected two out of the three previously mentioned TTA TSF methods for comparison?

---

> > > > > ### Author Response · Authors · 2025-11-28
> > > > >
> > > > > In our previous evaluation of the PETSA method, we occasionally observed outputs containing NaN values during hyperparameter tuning, which we suspect may be due to instability arising from test-time updates. Additionally, the performance of PETSA was found to be generally comparable to the other two TTA methods. For completeness, we have now included all results as follows.
> > > > >
> > > > > |     Model⬇      |  dataset⬇   | TATO_mse_imp | DynaTTA_mse_imp | TAFAS_mse_imp | PETSA_mse_imp |
> > > > > | :-------------: | :---------: | :----------: | :-------------: | :-----------: | :-----------: |
> > > > > | **Timer-LOTSA** |    ETTh1    |   **7.3%**   |      -8.2%      |     -0.3%     |     0.0%      |
> > > > > |                 | Electricity |     0.0%     |      -1.8%      |   **0.2%**    |     0.0%      |
> > > > > | **Timer-UTSD**  |    ETTh1    |   **5.1%**   |     -18.2%      |     0.0%      |     0.0%      |
> > > > > |                 | Electricity |   **0.8%**   |      -3.7%      |     0.0%      |     -0.7%     |

---

### Official Review · Reviewer_69Mi · 2025-11-01

**Soundness:** 2
**Presentation:** 2
**Contribution:** 2
**Rating:** 2
**Confidence:** 4

**Summary:**

The paper is flawed due to deficient writing, a lack of crucial experimental comparisons against a fine-tuned LTM and other established time-series methods, and the absence of a required quantitative analysis on computational overhead.

**Strengths:**

The authors provide sufficient ablation studies and comparisons, which enhance the credibility of the findings.

**Weaknesses:**

1. The writing is deficient, like the transition from paragraph in line 55 to 57 feels abrupt; Section 2.3 is a seemingly extraneous point, confusing the reader and disrupting the narrative flow; Section 3.1 is repetitive because the same content has been discussed before; In figure 2, it seems like the order of 3 types of operators is reorderable, but it's said in the text that they are fixed, it's confusing which ones are reorderable; In Section 3.2.2, post-process operators are mentioned, but lacks a explaination of what exactly they are; Section 4.2 is repetitive too.
2. You have verified that your method yields better results than the baseline (without TATO) on a specific dataset. However, how does it compare to LTM that has been fine-tuned on the same dataset? A sole comparison between having TATO and not having TATO is insufficient to prove the efficacy of your method, as direct fine-tuning on the LTM might achieve superior results.
Furthermore, regarding the SOTA method, how much computational resource saving does it actually provide? You need to compare the overhead and computational resources required by your method against those needed for fine-tuning the LTM, rather than merely stating how little time your method requires.
3. As I understand it, TATO is merely a combination of several common transformation methods. Given this, there are many other established methods for processing time-series data—is your approach superior to them?
If your claimed advantage is superior generalization, then you must also provide evidence that other methods perform worse than yours in terms of generalization. It is not enough to simply state that these other methods are designed for "specific problems"; this claim needs to be supported by experiments.
4. I don't understand the point of section 4.5. What specific benefit of your method are you trying to demonstrate with this experiment? Furthermore, why are only two models used in this experiment? What about the others?
"First, we finetune Timer-UTSD over all eight datasets above to get a single LTM that works well on all of them." How well does it work? Lacks experimental data.

**Questions:**

No

---

> ### Author Response · Authors · 2025-11-23
> **Weakness 1**
>
> We sincerely thank Reviewer 69Mi for providing a detailed review and insightful questions.
>
> > **W1(1):** "Some issues on presentations"
>
> We respond to each question on presentations as follows:
>
> - The paragraph at line 51 summarizes our motivation. The following paragraph, at line 57, explains the novelty and significance of this motivation. We do not see any noteworthy issues with this demonstration.
>
> - Section 2.3 is an independent section of related works. The proposed method is used to modify the performance of a frozen model, which is also supported by test-time adaptation methods. We think it is necessary to present a comparison of such related works for readers. And the "narrative flow" seems to be OK with this.
>
> - Section 3.1 is the first part of our method demonstration, and we start with two definitions to present the proposed paradigm of adapting data to model. The first three sentences introduce the paradigm with the issues the LTMs are facing. The issues are also presented differently at line 46 in the introduction section. Such echoing is commonly seen in research papers to be friendly to readers. We think it is inappropriate to treat it as a drawback of a paper.
>
> - Section 4.2 describes how we evaluate the proposed method. The description is not repetitive with anything in the main body of the paper. Maybe Reviewer 69Mi mistook this for Section 3.2.3, which is part of our method for evaluating the trials during adaptation.
>
> > **W1(2):** "Missing presentations"
>
> - For reorderable transformations. Figure 2 clearly shows that the transformations are reorderable. We also present this in detail in the paragraph at line 258. The transformations are reordered using several heuristic rules because many meaningless orders are improper and reduce the optimization's efficiency.
>
> - For post-processing. We provide detailed definitions of the transformations in Appendix 3. The post-processes restore the original value scope and are straightforward. For transformations that change the value scope, such as scaler, differencer, and warper, a restore computation is needed, so post-processing, such as denormalization, is performed. For transformations that do not change the value scope, such as trimmer, aligner, denoiser, and imputater, there is no post-process. We will supplement the details in the revised version.

---

> ### Author Response · Authors · 2025-11-23
> **Weakness 2**
>
> > **W2:** "Comparison with finetuning on single datasets"
>
> Thank you for the insightful question.
>
> We conduct an additional comparison between TATO and finetuning on each dataset to provide a more comprehensive understanding of TATO's efficacy.
>
> In finetuning experiments, we use the default implementation from "Timer: Transformers for Time Series Analysis at Scale" and keep the finetune settings and hyperparameters aligned with their official code release. The samples used for finetuning are the same as those needed by TATO for fairness (500 samples per dataset). The finetuned checkpoints are applied to the same testing logic aligned to TATO.
>
> - Effectiveness:
>     - **In most cases, TATO reaches similar or better performance compared to finetuning (on each dataset)**.
>     - For smaller datasets such as ETT and Exchange, TATO consistently outperforms finetuning.
>     - For larger datasets such as Traffic, Weather, and Electricity, under a higher diversity of the samples, the advantage of TATO reduces, but it still achieves similar results.
> - Efficiency:
>   - Time cost: The timespan of the finetuning process is highly unstable due to different dataloading parallel techniques and dataset size. In our experiment, fine-tuning Timer typically takes about 300 seconds. TATO with 500 trials takes 76 seconds on Timer, which is about **25\% of the finetuning timespan**.
>   - GPU memory: The finetuning of Timer requires at least 4GB of GPU memory, while TATO on Timer only requires 1GB, just **25\% of the finetuning GPU memory**.
>
> | MSE/MAE     | Timer-UTSD-finetune | Timer-UTSD-TATO | Timer-LOTSA-finetune | Timer-LOTSA-TATO |
> | ----------- | ------------- | --------------- | -------------- | ---------------- |
> | Electricity | 0.177/0.290   | **0.174/0.287**     | 0.1705/**0.287**   | **0.1703**/0.289     |
> | ETTh1       | 0.419/0.492   | **0.384/0.473**     | 0.443/0.506    | **0.411/0.484**      |
> | ETTh2       | 0.338/0.449   | **0.318/0.434**     | 0.382/0.487    | **0.352/0.463**      |
> | ETTm1       | 0.302/0.401   | **0.280/0.386**     | 0.377/0.461    | **0.248/0.356**      |
> | ETTm2       | 0.478/0.529   | **0.442/0.488**     | 0.317/0.420    | **0.249/0.352**      |
> | Exchange    | 0.522/0.499   | **0.323/0.397**     | 0.846/0.712    | **0.231/0.344**      |
> | Traffic     | **0.058/0.132**   | 0.059/0.135     | 0.0615/**0.138**   | **0.0614**/0.140     |
> | Weather     | **0.519/0.508**   | 0.604/0.527     | 0.608/0.562    | **0.593/0.542**      |
> | **AVERAGE**     | 0.352/0.413   | **0.323/0.391**     | 0.400/0.447    | **0.289/0.371**      |
>
>
> Furthermore, we conduct additional experiments to show how TATO can improve finetuning. We use TATO to select a set of data transformations, then apply them for finetuning.
>
> | Dataset | Setting  (input and output) | vanilla Timer-UTSD (MSE/MAE） | Timer-UTSD+TATO (MSE/MAE) |
> | ------- | ------- | --------------------------------- | ---------------------------- |
> | ETTh1   | 672-24  | **0.0319/0.1352**                     | 0.0333/0.1414                |
> | ETTh1   | 672-48  | 0.0530/0.1787                     | **0.0465/0.1693**                |
> | ETTh1   | 672-96  | 0.0776/0.2228                     | **0.0547/0.1821**                |
> | ETTh1   | 672-192 | 0.0815/0.2302                     | **0.0747/0.2151**               |
> | Traffic | 672-24  | 0.0981/0.1805                     | **0.0884/0.1649**                |
> | Traffic | 672-48  | 0.1017/0.1849                     | **0.0985/0.1719**                |
> | Traffic | 672-96  | 0.1053/0.1809                     | **0.1033/0.1741**                |
> | Traffic | 672-192 | 0.1341/0.2152                     | **0.1336/0.2079**                |
>
> The experiments demonstrate that TATO can provide a better strategy for data transformations, offering a new way to use TATO.

---

> ### Author Response · Authors · 2025-11-23
> **Weakness 3 and 4**
>
> > **W3:** "The advantages of the proposed paradigm over specific transformations"
>
> **TATO is a paradigm of "data optimization for model" rather than a naive combination of common transformations**. The optimal transformations for different datasets are different according to their characteristics. TATO stands for a novel approach to improve the performance of LTMs by using dynamic transformations rather than predefined ones.
>
> Test-time adaptation methods also allow data to be adapted under a specific assumption about data characteristics. We conduct additional experiments comparing with test-time adaptation. **Unlike TATO, existing test-time adaptation methods mainly address forecasting issues for conventional deep learning models rather than LTMs**. Under consistent experimental settings with TATO, we evaluated two test-time adaptation approaches (DynaTTA and TAFAS) for Timer-LOTSA and Timer-UTSD models on the datasets (ETTh1 and Electricity).
>
> In general, **TATO achieves more substantial performance improvements and demonstrates greater data-adaptation capability**. However, both test-time adaptation approaches perform poorly across the two LTMs. We will conduct more experiments with comprehensive comparisons to test-time adaptation in the final version of the paper.
>
> |     Model     | Dataset | TATO_mse_imp | DynaTTA_mse_imp | TAFAS_mse_imp |
> | :-------------: | :------: | :----------: | :-------------: | :-----------: |
> | **Timer-LOTSA** |  ETTh1   |   **7.3%**   |      -8.2%      |     -0.3%     |
> |                 |   ECL    |     0.0%     |      -1.8%      |   **0.2%**    |
> | **Timer-UTSD**  |  ETTh1   |   **5.1%**   |     -18.2%      |     0.0%      |
> |                 |   ECL    |   **0.8%**   |      -3.7%      |     0.0%      |
>
>
> > **W4:** "Section 4.5"
>
> The primary focus of this paper is on enhancing the zero-shot performance of foundational models through data transformations. The experiments presented in section 4.5 are supplementary to our mainstream experiments, aiming to provide insight into the question: "Can TATO still be useful for cross-domain finetuned models?"
>
> In Section 4.5, we provide an additional experiment on finetuned foundational models. Specifically, **we select 500 samples (the same number as in TATO) from each dataset, concatenate these subsets into a joint cross-domain dataset, and fine-tune Timer on it**. We found that TATO can still deliver performance gains after cross-domain finetuning, demonstrating its effectiveness for cross-domain finetuned LTMs. Technically, **the cross-domain finetuning setting here aligns with FrozenForecasting defined in the paper**. In contrast, finetuning on a single dataset breaks the FrozenForecasting setting.
>
> The experiments in Section 4.5 are used to provide additional insights rather than a primary result. Only two models are presented in Section 4.5, i.e., Timer-UTSD and Timer-LOTSA. This is because among the LTMs we have included in our discussion, only Timer provides standard finetuning APIs. To avoid unnecessary arguments about the finetuning process, we prefer to use the finetuning methods offered by the original LTMs' authors.
>
> Thanks again for the questions.

---

### Author Response · Authors · 2025-11-29
**A summary of the rebuttal**

Dear Area Chair and Reviewers,

To better understand the rebuttal of our work, we provide a summary of the existing reviews and responses.

1. Reviewer Mentioned Primary Strengths:
- It is **a novel idea and original paradigm on aligning data with LTMs to improve domain-adaptive forecasting performance**.
- It implements a **well-designed and efficient pipeline**.
- Experiments provide **sufficient evidence of its significant performance and universal applicability across various LTMs**.

2. Reviewer Mentioned Primary Weaknesses, Questions, and Our Responses:
- Require additional **comparison with finetuning on single datasets**.
    - We conducted additional experiments comparing TATO with finetuning on single datasets and using TATO to further enhance finetuning, demonstrating TATO's broad applicability and high efficiency.
- Require additional **comparison with test-time adaptation methods**.
    - We conducted additional experiments that compare TATO with three test-time adaptation methods, which exhibit TATO's more stable effectiveness for LTMs.
- Require more **explanation on scalability and robustness**.
    - We further explained Figure 4 to show our contribution to robustness.
    - We conducted additional experiments with a larger scale of settings, which exhibit TATO's robustness and high efficiency.
- Require deeper **analysis on the advantages** of the work.
    - We provided the distribution of the selected transformations over different datasets and models, providing a new perspective on analysing the gap between data and model.


3. Key Summary by Each Reviewer:
- Reviewer 69Mi
    - Reviewer 69Mi was negative with a score of 2, and there is still no response to our rebuttal.
    - We want to point out that, as we commented following the response, we think **the obscure and disproportionate comments on minor presentational aspects should not outweigh the novel findings and rigorous methodology** presented in our paper.
 - Reviewer YMH3
     - Reviewer YMH3 has become positive after rebuttal, and **claims to raise the score, i.e., the score should have changed from 4 to 6 or 8**.
 - Reviewer kQ8y
     - Reviewer kQ8y has always been positive, and **has raised the score from 6 to 8 after our rebuttal**.
 - Reviewer 8T1Q
     - Reviewer 8T1Q was positive with **a score of 6**, and there is still no response to our rebuttal.

**We claim that none of the authors in this paper attempted to contact any reviewer during the submission or rebuttal process**. From the current responses to the rebuttal, we can see that:
- The questions proposed by the reviewers primarily focus on additional comparison with other methods, for which **we provided sufficient responses with comprehensive experiments**.
- From the existing responses to our rebuttal, **the reviewers expressed their strong acknowledgement with claims of raising the scores**.
- At least three reviewers acknowledge the novelty, applicability, scalability, and robustness of our work, and they agree to accept our paper.
- We have no way to know the latest attitude of Reviewer 69Mi to our work, but from the two primary questions about comparison with finetuning and test-time adaptation, which Reviewer YMH3 and Reviewer kQ8y also asked, **we have good reason to believe that Reviewer 69Mi would be positive to our work**.

---

### Meta-Review · Area_Chair_Yfnr · 2026-01-07

**Summary:**

This paper proposes TATO, a novel data-centric adaptation paradigm for domain-shared time series foundation models (TSFMs), shifting the focus from model adaptation to adaptive transformation optimization of input data. By automatically searching for empirically optimal transformation pipelines, including context slicing, normalization, and outlier correction, TATO enables frozen TSFMs to better adapt to diverse target domains without fine-tuning. The proposed two-stage ranking mechanism further improves robustness and stability across metrics. The problem is well motivated, and the paradigm is original and practically appealing, especially in scenarios where fine-tuning is costly or infeasible.

Reviewers generally recognized the novelty of the “adapt data to model” perspective, the efficiency and practicality of the pipeline, and the strong empirical performance across multiple state-of-the-art TSFMs and standard forecasting benchmarks. Importantly, the reported gains are often substantial while incurring minimal computational overhead, making the approach attractive for real-world deployment.

The rebuttal was particularly strong and constructive. The authors addressed the main concerns raised by reviewers by adding extensive new experiments and analyses, including direct comparisons with single-dataset fine-tuning, test-time adaptation methods, scalability studies, sampling sensitivity analyses, and additional results on non-LTM models such as PatchTST. These additions significantly strengthened the paper and clarified the scope, robustness, and efficiency–effectiveness tradeoffs of the proposed framework. Multiple reviewers explicitly acknowledged the rebuttal and raised their scores accordingly.

At the same time, some limitations remain. The framework is primarily empirical, with limited theoretical grounding on why certain transformations generalize across domains, and the transformation search space is manually defined, which may restrict adaptability in highly specialized settings. While these aspects do not undermine the validity of the contribution, they suggest that the work is more practical and engineering-driven than conceptually transformative.

Overall, TATO represents a solid, original, and practically useful contribution to the adaptation of time series foundation models. Given its strengths and remaining limitations, acceptance is the most appropriate outcome.

**Reviewer Concerns:**

Reviewers raised concerns regarding presentation clarity, missing comparisons with fine-tuning and test-time adaptation methods, scalability, sampling strategies, and theoretical justification. These concerns were largely addressed in the rebuttal through additional experiments, detailed analyses, and clarifications.

One reviewer (Reviewer 69Mi) assigned a low score; however, this reviewer did not participate in the post-rebuttal discussion and did not engage with the authors’ responses. As a result, the weight of this review is limited in assessing whether the rebuttal resolved the raised concerns.

Overall, after considering the full set of reviews and the rebuttal, the remaining issues are relatively minor and mainly relate to theoretical depth and extensibility.

**Reviewer Scores:**

Reviewer scores were initially mixed. During the rebuttal phase, multiple reviewers explicitly indicated score increases after their concerns were addressed, while others maintained positive or borderline-positive evaluations.

One reviewer (Reviewer 69Mi) assigned a score of 2 but did not participate in the discussion after the rebuttal and did not update their assessment. Given the lack of engagement with the rebuttal, this score is less informative for evaluating the final quality of the submission.

---

### Decision · Program_Chairs · 2026-01-26

Accept (Poster)